# 3D-Agent: A Tri-Modal Multi-Agent Responsive Framework for Comprehensive 3D Object Annotation

**Jusheng Zhang[1], Yijia Fan[1], Zimo Wen[2], Jian Wang[3], Keze Wang[1,†]**

[1] Sun Yat-sen University
[2] Shanghai Jiao Tong University
[3] Snap Inc.
[†] Corresponding author: `kezewang@gmail.com`

## Abstract

Driven by the applications in autonomous driving, robotics, and augmented reality, 3D object annotation is a critical task compared to 2D annotation, such as spatial complexity, occlusion, and viewpoint inconsistency. The existing methods relying on single models often struggle with these issues. In this paper, we introduce Tri-MARF, a novel framework that integrates tri-modal inputs (i.e., 2D multi-view images, text descriptions, and 3D point clouds) with multi-agent collaboration to enhance the 3D annotation process. Our Tri-MARF consists of three specialized agents: a vision-language model agent that generates multi-view descriptions, an information aggregation agent that selects optimal descriptions, and a gating agent that aligns text descriptions with 3D geometries for more refined captioning. Extensive experiments on the Objaverse-LVIS, Objaverse-XL, and ABO datasets demonstrate the superiority of our Tri-MARF, which achieves a CLIPScore of 88.7 (compared to 78.6–82.4 for other SOTA methods), retrieval accuracy of 45.2/43.8 (ViLT R@5), and an impressive throughput of 12,000 objects per hour on a single NVIDIA A100 GPU.

## 1 Introduction

3D object annotation is crucial in computer vision, providing semantic labels for 3D data across autonomous driving [30, 5, 28, 24, 17, 40, 78, 61], robotics, and AR applications. The existing methods primarily use single large-scale models without leveraging multi-agent collaboration, creating challenges with complex scenes [26, 17, 20, 73, 52, 71]. Unlike 2D annotation, 3D annotation faces unique difficulties: increased spatial relationship complexity, occlusion issues [13, 1, 32, 70], and viewpoint variations affecting cross-view consistency [45]. Traditional 3D annotation methods face serious problems with multi-view data: single models struggle to handle viewpoint differences, geometric complexity, and semantic consistency simultaneously. When objects are partially occluded or only visible from specific angles, existing methods often generate incomplete or inconsistent annotations. Existing approaches, which typically rely on vision-language models, often encounter hallucination problems and description inconsistencies [75, 58, 67]. The existing methods struggle to maintain perspective consistency when using multi-view information and often overlook inherent geometric information by relying solely on 2D images [42, 14].

After performing deep and comprehensive analysis of these challenges, we find that it is vital to overcome the inherent difficulty of a single decision-making system by optimizing multiple competing objectives simultaneously, i.e., accuracy, completeness, consistency [64, 46, 41], and efficiency [18]. In complex 3D annotation tasks, a single model often struggles to balance these goals effectively [62], akin to a solitary expert lacking proficiency across all domains. These formidable challenges inspire a key question: how can we design a system that collaborates like a team of human experts,

39th Conference on Neural Information Processing Systems (NeurIPS 2025).

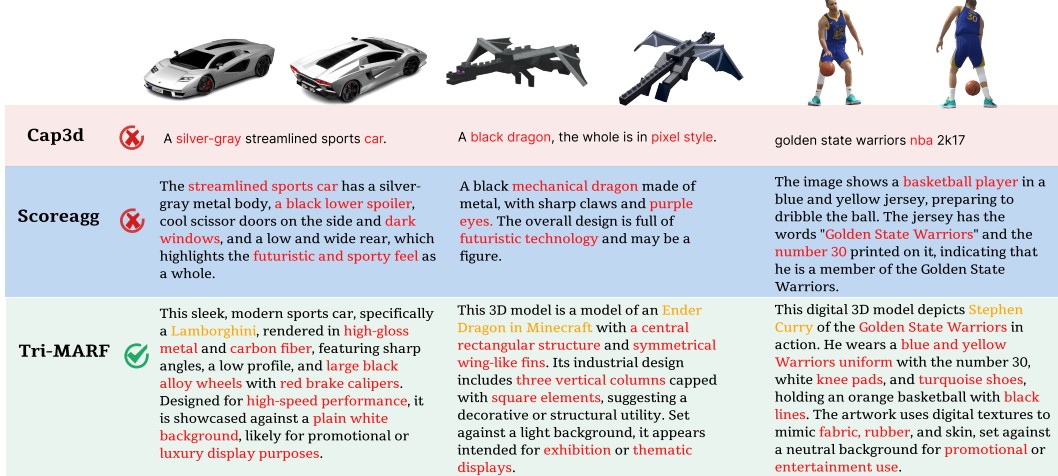

| | | | | |
|---|---|---|---|---|
| **Cap3d** | ✗ | A silver-gray streamlined sports car. | A black dragon, the whole is in pixel style. | golden state warriors nba 2k17 |
| **Scoreagg** | ✗ | The streamlined sports car has a silver-gray metal body, a black lower spoiler, cool scissor doors on the side and dark windows, and a low and wide rear, which highlights the futuristic and sporty feel as a whole. | A black mechanical dragon made of metal, with sharp claws and purple eyes. The overall design is full of futuristic technology and may be a figure. | The image shows a basketball player in a blue and yellow jersey, preparing to dribble the ball. The jersey has the words "Golden State Warriors" and the number 30 printed on it, indicating that he is a member of the Golden State Warriors. |
| **Tri-MARF** | ✓ | This sleek, modern sports car, specifically a Lamborghini, rendered in high-gloss metal and carbon fiber, featuring sharp angles, a low profile, and large black alloy wheels with red brake calipers. Designed for high-speed performance, it is showcased against a plain white background, likely for promotional or luxury display purposes. | This 3D model is a model of an Ender Dragon in Minecraft with a central rectangular structure and symmetrical wing-like fins. Its industrial design includes three vertical columns capped with square elements, suggesting a decorative or structural utility. Set against a light background, it appears intended for exhibition or thematic displays. | This digital 3D model depicts Stephen Curry of the Golden State Warriors in action. He wears a blue and yellow Warriors uniform with the number 30, white knee pads, and turquoise shoes, holding an orange basketball with black lines. The artwork uses digital textures to mimic fabric, rubber, and skin, set against a neutral background for promotional or entertainment use. |

Figure 1: Comparison example of our Tri-MARF captions with previous SOTA methods. Our Tri-MARF not only accurately recognizes the specific names of objects, but also provides rich and correct details. Some keywords in the annotations are shown in red, and the specific names of the objects are shown in orange. Please note that only our Tri-MARF can mark them out.

addressing the task from multiple specialized perspectives [49, 55]. Motivated by this, multi-agent systems offer a natural framework by decomposing complex tasks into specialized sub-tasks, allowing distinct agents to leverage their respective strengths. For 3D object annotation, this implies deploying dedicated agents to handle geometric feature recognition, semantic understanding, and cross-view consistency separately. However, a central challenge in multi-agent systems lies in effectively coordinating the decisions of diverse agents [31, 66], particularly under uncertainty and conflicting information. To address this coordination problem, reinforcement learning emerges as an ideal solution. By dynamically learning optimal policies, reinforcement learning techniques [43, 44] enable the system to continuously refine and optimize collaborative decision-making, surpassing the limitations of predefined rules. Integrating multi-agent systems with reinforcement learning offers numerous advantages [50, 35, 57, 68], including robustness, adaptability, and enhanced performance in tackling complex problems [36, 76]. However, this introduces new difficulty, e.g., designing appropriate reward signals to evaluate annotation quality and embedding reinforcement learning seamlessly into the workflow.

To address this issue, we propose a novel annotation framework called Tri-MARF (Tri-Modal Multi-Agent Response Framework). The core idea is to adopt a multi-stage pipeline where specialized agents handle each task, with reinforcement learning incorporated to enhance decision-making, particularly in the critical text aggregation phase. As shown in Figure 2, our Tri-MARF consists of four stages to progressively refine and integrate annotation information: **Data Preparation Stage**: 3D objects are sourced from datasets like Objaverse [12, 11], generating multi-view 2D images and point cloud features to capture structural details; **Initial VLM Annotation Stage**: A vision-language model (VLM) agent generates preliminary descriptions for each viewpoint. To ensure accuracy, we employ multi-round Q&A with Qwen2.5 [39]. For each view, five candidate responses are produced, and a RoBERT [23] model clusters these using DBSCAN to yield the text description for that perspective; **Reinforcement Learning-Based Information Aggregation Stage**: We introduce an agent based on a multi-armed bandit [48] with an upper confidence bound algorithm to aggregate candidate descriptions from different views into a coherent, high-confidence global description. This agent models multi-view annotation as a multi-armed bandit problem, where each description candidate serves as an "arm" and the system dynamically learns to select optimal descriptions through exploration-exploitation balance. Unlike static rules or simple voting, our agent adapts to different object types and viewpoint scenarios. This agent learns to balance visual consistency, geometric accuracy, and semantic richness through reward functions. This agent is trained using a composite reward function incorporating VLM confidence scores and CLIP [40] similarity between images and generated captions to dynamically balance exploration and exploitation and optimizing cross-view description consistency through continuous learning; **Gating Stage**: Cosine similarity between aggregated text and 3D point cloud is computed via an encoder [53], with a threshold determining

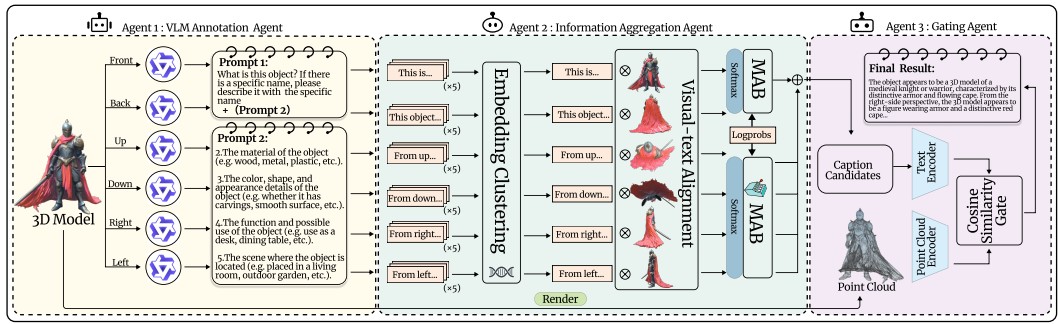

Figure 2: The illustration of our Tri-MARF for 3D object annotation, featuring a collaborative multi-agent mechanism. The process starts with Agent 1 (VLM Annotation Agent), which uses a visual language model (e.g., Qwen2.5-VL-72B-Instruct) to generate 5 text descriptions for each view of a 3D object from six standard viewpoints (front, back, left, right, top, bottom). These descriptions are then processed by Agent 2 (Information Aggregation Agent), which uses RoBERTa+DBSCAN for semantic embedding clustering, CLIP for visual-text alignment, and integrates a multi-armed bandit (MAB) model to optimize description selection and balance exploration and exploitation to obtain the final captions. Agent 3 (Point Cloud Gating Agent) uses threshold control to align text and 3D point clouds, further reducing the wrong results produced by VLM annotations. Please note that our point cloud is a pre-rendered asset.

further annotation needs. Adaptive agent outperforms rule-based methods by dynamically adapting to scenes and learning from errors to optimize 3D annotation quality (Figure 1).

This **main**paper introduces Tri-MARF, a novel multi-stage annotation framework that integrates specialized agents to tackle inconsistencies in 3D object annotation. By leveraging multi-agent collaboration with reinforcement learning, Tri-MARF enhances decision-making and ensures annotation consistency, offering a robust and adaptive solution. Extensive experiments on benchmarks like Objaverse-LVIS, Objaverse-XL, and ABO demonstrate Tri-MARF's superior performance, surpassing existing methods in annotation accuracy, description consistency, and linguistic quality. To date, it has annotated approximately 2 million 3D models.

## 2 Related Works

**Neural 3D Object Annotation** has evolved from manual labeling to automated approaches. Chang et al. [7] established ShapeNet, Mo et al. [29] developed PartNet, Yi et al. [63] focused on semantic segmentation, and Savva et al. [38, 60] contributed SHREC benchmarks. Recent vision-language models transformed this landscape: ULIP [58] bridged 3D point clouds and text, PointCLIP [75] adapted CLIP for point clouds, while [45] introduced cross-modal embeddings and Zeng et al. [51] explored prompt engineering. Cap3D [26] pioneered synthetic-to-real transfer but struggled with cross-view consistency. 3D-LLM [17] attempts to solve this via specialized objectives, while [32] developed a complex scene understanding method.

**Multi-Agent Systems for Visual Understanding** decompose complex visual tasks into manageable subtasks to enhance efficiency and accuracy in visual understanding. Maes et al. [27] established the foundational concept of collaborating agents, Zhong et al. [77] demonstrated that specialists outperform monolithic models, and Deng et al. [13] introduced multi-agent approaches to 3D scene understanding. Deng et al. [47] and Zhong et al. [77] showed improved robustness through viewpoint integration, while Aghasian et al. [1] developed hierarchical protocols for agent collaboration, and Chafii et al. [4] implemented emergent communication between agents. For scene comprehension, Wei et al. [54] proposed 3D scene graph generation frameworks, Johnson et al. [19]'s DenseCap system demonstrats the importance of specialized roles for image annotation, and Cai et al. [3] and Liu et al. [22] explored dynamic agent routing to enhance system adaptability. Unlike these fixed-protocol systems, our Tri-MARF introduces a tri-modal approach with multi-agent collaboration to optimize collaboration, adaptively weighting information across different modalities to maintain high performance across varying scene complexities.

**Reinforcement Learning for Decision Making in Vision Systems** has employed reinforcement learning to optimize viewpoint selection in 3D environments [30], complemented by advances in exploration strategies that improve training efficiency for visual perception tasks [65]. Recently, multi-armed bandit algorithms [48], including upper confidence bound strategies that balance exploration and exploitation [2], have been well researched. These approaches have proven effective for content selection in visual applications [79] and have been extended to contextual settings where visual representations inform decision-making [8]. Multi-modal fusion research has advanced through adaptive weighting mechanisms based on reinforcement learning principles [16], particularly in vision-language tasks where modality importance varies contextually [15, 69]. Recent advances [72, 74] demonstrate the ongoing evolution of these approaches for complex decision-making tasks. In contrast, our Tri-MARF implements a simple yet effective architecture with specialized agents for 3D annotation tasks, demonstrating superior adaptability across diverse object categories and viewpoint conditions.

## 3 Methodology

Our Tri-MARF framework addresses key challenges in 3D annotation through three specialized agents in the following four-stage annotation pipeline. Formally, suppose $V = \{front, back, left, right, top, bottom\}$ represent the standardized viewpoints. **Data Preparation Stage** generates six corresponding images $\{I_v : v \in V\}$ for each object, which are encoded and passed to the vision-language model without manual feature engineering. For each viewpoint $v$, the **Initial VLM Annotation Stage** produces: $D_v = \{C_{v,i}\}_{i=1}^{M}$, a set of $M$ candidate descriptions (Tri-MARF employs $M = 5$ with temperature-controlled sampling) each with an associated confidence score. **Information Aggregation Stage** transforms these texts into BERT embedding space for semantic clustering, while CLIP evaluates each description's visual alignment with image $I_v$. This dual-evaluation process produces scored response pairs $(C_{v,i}, s_{v,i})$, where $s_{v,i}$ represents a composite score of semantic distinctiveness and vision-text correlation. In **Information Aggregation Stage**, we frames candidate descriptions as arms in a multi-armed bandit problem, selecting one description $\hat{C}_v$ per view through UCB (Upper Confidence Bound) exploration-exploitation balancing. The information aggregation agent continuously updates reward estimates to favor higher-quality descriptions as it learns. Finally, we fuse the optimized view-specific descriptions $\hat{C}_v : v \in V$ into a coherent global annotation, with weighted emphasis on informative perspectives. **Gating Stage** input the description obtained in the previous step into the pretrained text encoder for encoding. At the same time, we input the 3D point cloud of the model obtained in the preparation stage into the pre-trained point cloud encoder for encoding. Then, we calculate the cosine similarity between the encoded text and the point cloud to determine whether the cosine similarity is greater than the empirical threshold $\alpha = 0.557$ (Please refer to Supp. 11.6). If it is greater than this threshold, the corresponding sample is retained; if it is less than this threshold, the sample is marked as a questionable sample for manual annotation.

### 3.1 Initial VLM Annotation Stage

For each view image $I_v$, Tri-MARF employs a sophisticated vision-language model agent that uses an innovative multi-turn prompting strategy. Unlike conventional single-prompt approaches, we implement a structured dialogue system with Qwen2.5-VL-72B-Instruct[39] that mirrors expert visual analysis. Our prompting protocol unfolds in three strategic phases: **Viewpoint-aware identification.** We orient the model to recognize its viewing perspective (e.g., "This is the front view. What object do you see and what is its specific name?"), ensuring attention focuses on viewpoint-specific diagnostic cues. **Systematic attribute elicitation.** We use targeted follow-up prompts to elicit key attributes such as color, material, and structural components, guaranteeing sufficient feature coverage even under complex viewpoints. **Contextual integration.** The extracted observations are integrated into consistent, coherent descriptions that preserve viewpoint alignment and emphasize distinguishing characteristics. This transforms annotation quality by decomposing complex visual reasoning into manageable sub-tasks, yielding significantly more detailed and accurate descriptions than conventional single-prompt methods. To maximize semantic coverage and reduce annotation bias, we empirically introduce stochastic diversity sampling at temperature $= 0.7$ with $M = 5$ descriptions per view, generating alternative interpretations that capture different object aspects. Each description $C_{v,i}$ retains token-level log-probabilities, enabling sophisticated confidence assessment.

**Confidence Score Computation**. We introduce a novel probabilistic confidence metric for each description, addressing the critical challenge of uncertainty quantification in 3D annotation. The confidence score $\text{Conf}(C)$ quantifies semantic reliability through average token log-likelihood, providing a principled measure of model certainty. For tokens $t_1, t_2, \ldots, t_N$ with conditional probabilities $P(t_i \mid \text{context})$, we compute:

$$\text{Conf}(C) \;=\; \frac{1}{N} \sum_{i=1}^{N} \big|\log P(t_i \mid \text{context up to } t_i)\big|. \tag{1}$$

This formulation captures the model's internal uncertainty during generation—lower $\text{Conf}(C)$ values indicate higher confidence (higher token probabilities), while elevated scores signal potential unreliability (perhaps from rare descriptors or uncertain attributions). This confidence metric serves dual purposes in our reinforcement learning pipeline: flagging potentially hallucinated content for rejection, and informing bandit-based selection between semantically similar candidate descriptions. This probabilistic approach to confidence assessment represents a significant advancement over deterministic methods, enabling more reliable annotation in ambiguous situations.

**Importance of Multi-View Inputs.** Our Tri-MARF innovatively processes six standard views of each 3D object, addressing the fundamental challenge of viewpoint inconsistency that plagues single-view methods. Unlike traditional approaches that rely on limited perspectives, our Tri-MARF captures comprehensive spatial relationships through front, back, left, right, top, and bottom views. This deliberate design tackles the inherent difficulty of 3D annotation—objects often conceal critical features from any single viewpoint. For instance, a vehicle's diagnostic features distribute across multiple angles: brand identifiers on the front, distinctive lighting arrays at the rear, profile silhouettes from sides, and functional components from top/bottom perspectives. Our Tri-MARF approach systematically mitigates occlusion problems by exploiting complementary information across perspectives, creating an integrated understanding impossible with conventional methods. This redundancy provides crucial resilience: when noise or occlusion compromises one viewpoint, alternative angles maintain annotation integrity. Furthermore, our Tri-MARF supports cross-view verification, confirming the existence of features across multiple viewpoints and reducing the inconsistency problem prevalent in standard VLM methods. This comprehensive spatial coverage forms the foundation for Tri-MARF's exceptional accuracy and completeness.

## 3.2 Information Aggregation Stage

After obtaining multiple description candidates per view, our Tri-MARF employs the aggregation agent to perform *semantic clustering* to eliminate redundancy and then *relevance weighting* to evaluate each description $C_{v,1}, \ldots, C_{v,M}$. These steps transform a raw list of $M$ descriptions into a smaller set of unique, scored responses, setting the stage for the final selection. To identify when different generated sentences are essentially saying the same thing, we project each candidate description into a high-dimensional semantic space using a pre-trained language model (BERT). Let $C_{v,i}$ and $C_{v,j}$ be two candidate descriptions for the same view $v$. We compute their embeddings $E_{v,i} = \text{BERT}(C_{v,i})$, $E_{v,j} = \text{BERT}(C_{v,j})$ as fixed-length vectors. The semantic similarity between the two descriptions is cosine similarity of their embedding vectors:

$$S_{ij} \;=\; \cos\big(E_{v,i}, E_{v,j}\big) \;=\; \frac{E_{v,i} \cdot E_{v,j}}{\|E_{v,i}\| \, \|E_{v,j}\|}. \tag{2}$$

performing this step, Tri-MARF condenses the candidate descriptions, removing duplicative entries and preparing a *canonical description* for each distinct idea. From each cluster, we select a representative description. Ideally, all descriptions in one cluster are paraphrases, so any could serve as the cluster's exemplar. Tri-MARF chooses the *highest-scoring* description in each cluster as the canonical representative. At this stage, we have not yet defined the score that comes next with the CLIP weighting, but once scores are assigned, we compute: $C^{(k)}\text{canonical} := \arg\max C_{v,i} \in \mathcal{C}ksv, i$, which produces a set of unique descriptions for the view one per cluster. These are the candidate descriptions for competing in the final selection.

### 3.2.1 Relevance Weighting

Next, our Tri-MARF evaluates how well each candidate description is grounded in the actual image using CLIP[40]. CLIP provides a function that maps an image $I$ and a text $T$ into a shared feature space such that related image-text pairs have high cosine similarity. We obtain an image embedding

$\mathbf{I}_v = f_{\text{CLIP}}^{\text{img}}(I_v)$ for the view $v$ and a text embedding $\mathbf{T}_{v,i} = f_{\text{CLIP}}^{\text{text}}(C_{v,i})$. The relevance of description $C_{v,i}$ to image $I_v$ is: $\cos\theta_{v,i}; =; \frac{\mathbf{I}_v \cdot \mathbf{T}_{v,i}}{|\mathbf{I}_v|;|\mathbf{T}_{v,i}|!}$ We convert the raw similarity into a probabilistic weight via a softmax over the $M$ descriptions of that view:

$$w_{v,i} \;=\; \frac{\exp\big(\cos\theta_{v,i}\big)}{\sum_{k=1}^{M}\exp\big(\cos\theta_{v,k}\big)}. \tag{3}$$

The candidate descriptions that align better with the image are assigned a higher weight. Then, our Tri-MARF combines the semantic clustering information and the CLIP visual alignment into a single score for each description. Let $S_{\text{conf},i}$ denote a confidence score for candidate description $i$, and let $w_i$ be the CLIP-based weight after softmax normalization. The final weighted score is: $s_i := (1-\alpha)\cdot S_{\text{conf},i} + \alpha\cdot w_i$, where $\alpha \in [0,1]$ controls the balance between text similarity and image-text similarity. In Tri-MARF, a smaller $\alpha$ prioritizes text-based confidence, while a larger $\alpha$ emphasizes visual-semantic alignment, combining signals for responses with both textual relevance and visual correctness.

### 3.2.2 Multi-Armed Bandit-Based Response Aggregation

Even after clustering and scoring, there may be multiple plausible descriptions for a given view. Rather than arbitrarily picking the top one, the information aggregation agent of our Tri-MARF uses a *multi-armed bandit (MAB)* model to adaptively select the best description over time, especially when feedback signals are available. Define the set of arms $\mathcal{A} = a_1, a_2, \ldots, a_K$ corresponding to the $K$ canonical descriptions for the current view. When Tri-MARF chooses arm $a_k$ (*i.e.*, uses description $C_{\text{canonical}}^{(k)}$ as the annotation), it receives a reward $r_k$ that reflects the annotation quality. Over many trials, the goal is to maximize:$\max_\pi;; \mathbb{E}\left[\sum_{t=1}^{T} r_{a_t}\right]$.We assume each arm $a_k$ has an underlying expected reward $\mu_k$. The challenge is the classic exploration-exploitation trade-off: the algorithm should try different arms to learn their rewards but also exploit the best one found so far. Tri-MARF employs the UCB1 variant, which calculates for each arm $a$: $a_t; =; \arg\max_{a\in\mathcal{A}}\left(\hat{r}_a; +; c\sqrt{\frac{2\ln t}{n_a}}\right)$, where $\hat{r}_a$ is the empirical mean reward, $n_a$ is how many times arm $a$ has been chosen, $t$ is the current round, and $c$ is an exploration weight. This rule formalizes "optimism in the face of uncertainty," ensuring that arms with high potential or insufficient exploration are tried sufficiently. When a reward $r_a$ is observed, the empirical mean is updated by $\hat{r}_a \leftarrow \frac{(n_a-1)\cdot\hat{r}_a + r_a}{n_a},$. Over time, Tri-MARF converges to favoring the arm with the highest true reward. We chose UCB due to its simplicity, strong regret bounds, and easy interpretability.

**Cross-View Processing and Global Description Synthesis.** Once the final descriptions of the individual views are selected, we ensure consistency and fuse them into a global 3D object annotation.**Front/Back View Prioritization**. Front and back views receive priority, assuming they carry critical identifying information about category and appearance. We assign higher weight $w_{FB}$ to these descriptions: $\text{Priority}(C_{FB}) = w_{FB}\cdot\text{Score}(C_{FB})$. In practice, front/back descriptions identify the object while other views (side, top, bottom) provide supplementary details. **Core Description Extraction**.

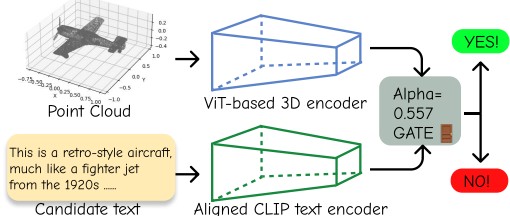
Figure 3: Detailed demonstration of the gating agent of our Tri-MARF. The pre-trained Uni3d encoder is used to handle point cloud and text matching on the open domain.

From the front/back combined description $C_{FB}$, Tri-MARF extracts only the first sentence as the core identification sentence: $S_{\text{core}} = \text{First\_Sentence}(C_{FB})$. For other views, a compiled description $C_{\text{other}}$ is formed by selecting the best or longest candidate from side/top/bottom views. **Global Description Assembly**. The final global description is then: $C_{\text{global}} = S_{\text{core}} + C_{\text{other}}$. A scoring formula like $\text{Score}_{\text{global}} = \frac{\text{Score}(C_{FB}) + \text{Score}(C_{\text{other}})}{2}$ evaluates how well the merged description aligns with both identity and detailed attributes.

Table 1: The comparison of caption quality and efficiency for 3D object annotation. The highest value of each metric is in bold. The two values of ViLT R@5 (e.g. 45.2/43.8) represent the retrieval performance of Image-to-Text (I2T) and Text-to-Image (T2I) respectively.

| Method | Objaverse-LVIS (1k) | | | Objaverse-XL (5k) | | | ABO (6.4k) | | | Speed (objects/hour) |
|---|---|---|---|---|---|---|---|---|---|---|
| | A/B Score | CLIPScore | ViLT R@5 | A/B Score | CLIPScore | ViLT R@5 | A/B Score | CLIPScore | ViLT R@5 | |
| Human Annotation | 2.3 | 82.4 | 40.0 / 38.5 | 2.9 | 81.0 | 37.0 / 35.5 | 2.9 | 78.9 | 33.8 / 32.5 | 0.12k |
| Our Tri-MARF | - | **88.7** | **45.2 / 43.8** | - | **86.1** | **40.5 / 38.9** | - | **82.3** | **37.1 / 35.6** | 12k |
| Cap3D | 3.3 | 78.6 | 35.2 / 33.4 | 3.5 | 76.4 | 32.1 / 30.5 | 3.5 | 74.8 | 28.9 / 27.3 | 8k |
| ScoreAgg | 3.9 | 80.1 | 37.8 / 36.0 | 3.7 | 78.5 | 34.5 / 33.0 | 4.2 | 76.2 | 31.2 / 30.0 | 9k |
| 3D-LLM | 3.2 | 77.4 | 34.9 / 33.3 | 3.4 | 75.6 | 31.8 / 30.3 | 3.3 | 73.0 | 28.4 / 26.9 | 6.5k |
| PointCLIP | 2.0 | 65.3 | 22.4 / 20.8 | 2.3 | 63.1 | 19.5 / 18.0 | 2.2 | 60.7 | 17.2 / 15.7 | 5k |
| ULIP-2 | 3.0 | 75.2 | 33.1 / 31.5 | 3.2 | 73.8 | 29.7 / 28.2 | 3.1 | 71.4 | 26.5 / 25.0 | 7k |
| GPT4Point | 1.8 | 62.9 | 18.7 / 17.1 | 2.0 | 60.5 | 16.3 / 14.8 | 1.9 | 58.2 | 14.6 / 13.1 | 4k |
| Metadata | 1.5 | 65.2 | 20.1 / 18.7 | - | - | - | 2.1 | 61.5 | 16.3 / 15.0 | - |

## 3.3 Gating Stage

To mitigate limitations of traditional 2D image annotation in discriminating geometric properties, we introduce a similarity gating agent based on point cloud-text alignment, as shown in Figure 3. We employ pre-trained encoders $\mathbf{E}_p$ (3D point cloud) and $\mathbf{E}_t$ (text) to extract geometric and semantic features respectively, both in $\mathbb{R}^d$ dimension. Cross-modal matching is quantified using: Cosine Similarity $= \frac{\mathbf{E}_p \cdot \mathbf{E}_t}{|\mathbf{E}_p|_2 |\mathbf{E}_t|_2}$ where $\cdot$ denotes dot product and $| \cdot |_2$ represents L2 norm. Based on validation grid search, we set dynamic threshold $\alpha = 0.577$ as confidence criterion. When similarity falls below threshold, the gating agent triggers dual-check: critical category samples undergo manual review while redundant samples are filtered out. This geometric-semantic consistency gating effectively suppresses annotation hallucination in visual language models and better leverages intrinsic 3D object information.

# 4 Experiments

We rigorously evaluate our Tri-MARF across four experiments to validate its 3D understanding capabilities: (1) Caption quality is assessed on Objaverse-LVIS [12], Objaverse-XL [11], and ABO [9] via A/B testing against human annotations and automated metrics (CLIPScore, ViLT [21] retrieval). (2) Type inference accuracy on Objaverse-LVIS is compared against CAP3D [25], ScoreAgg [20], and Human Annotation using GPT-4o [33] scoring and human validation. (3) The effect of selecting different numbers of viewpoints on the annotation quality is used to justify why we choose 6 viewpoints instead of 8 viewpoints in the previous work [78]. (4) We also conducted annotation experiments on clean 3D point cloud datasets and real-world noisy datasets, demonstrating that our Tri-MARF has high generalization performance. Please refer to Supp. 14 for more detailed experimental settings.

## 4.1 3D Captioning Test

**Experimental Setup.** We evaluate the caption quality of our Tri-MARF for 3D object annotation on Objaverse-LVIS (1k sampled), Objaverse-XL (5k sampled), and ABO (6.4k objects). The captions of our Tri-MARF are compared with those from Cap3D, ScoreAgg, ULIP-2 [59], PointCLIP [75], 3D-LLM [17], GPT4Point [37], human annotations, and metadata in terms of quality and efficiency. Random sampling ensures representativeness. Quality is measured via A/B testing (1-5 scale), CLIPScore, and ViLT retrieval. Note that the speed here is estimated according to the overall rate. Baseline models follow official configurations. Tri-MARF's detailed settings are in Supp. 6. We also explored the model's ability to understand scenes (See Supp. 9) **Experimental Results and Analyses.** As shown in Table 1, our Tri-MARF achieves state-of-the-art performance across all semantic alignment metrics while maintaining the highest annotation throughput (12k objects/hour) on a single NVIDIA A100 GPU. By design, Tri-MARF serves as the reference baseline for human preference evaluation (A/B scores), implicitly outperforming all methods through pairwise comparisons. Tri-MARF dominates CLIPScore (88.7 vs. 78.6–82.4 on Objaverse-LVIS) and ViLT R@5 (45.2/43.8 vs. 35.2–40.0), demonstrating superior cross-modal alignment. Notably, Tri-MARF-generated captions surpass human annotations in semantic precision (CLIPScore +6.3 on ABO) while avoiding human annotators' preference bias. Please refer to Supp. 12 for more details. Tri-MARF excels in 3D object captioning, with higher CLIPScore and ViLT retrieval accuracy showing effective feature capture by the multi-agent approach. High A/B test scores confirm the reinforcement learning-based aggregated

Table 2: Cross-dataset generalization experimental results comparison. The highest values are highlighted in bold.

| Method | ShapeNet-Core | | | ScanNet | | | ModelNet40 | | |
|---|---|---|---|---|---|---|---|---|---|
| | CLIP↑ | ViLT R@5↑ | GPT-4↑ | CLIP↑ | ViLT R@5↑ | GPT-4↑ | CLIP↑ | ViLT R@5↑ | GPT-4↑ |
| Human Annotation | 81.7 | 37.8 / 36.0 | 4.2 | 79.5 | 34.8 / 33.2 | **4.3** | 80.2 | 36.0 / 34.5 | 4.0 |
| Our Tri-MARF | **83.2** | **38.6 / 36.8** | **4.3** | **80.3** | **35.2 / 33.7** | 4.0 | **81.5** | **36.7 / 35.2** | **4.2** |
| Cap3D | 76.5 | 33.1 / 31.5 | 3.6 | 73.2 | 29.8 / 28.1 | 3.2 | 74.3 | 31.2 / 29.8 | 3.4 |
| ScoreAgg | 79.1 | 35.4 / 33.9 | 3.9 | 75.6 | 32.1 / 30.4 | 3.5 | 77.2 | 33.8 / 32.3 | 3.7 |
| 3D-LLM | 75.8 | 32.5 / 30.9 | 3.5 | 72.5 | 29.1 / 27.6 | 3.1 | 73.6 | 30.7 / 29.2 | 3.3 |
| PointCLIP | 63.4 | 21.7 / 20.2 | 2.3 | 60.8 | 19.3 / 17.9 | 2.1 | 62.1 | 20.5 / 19.1 | 2.2 |
| ULIP-2 | 73.7 | 31.4 / 29.8 | 3.3 | 70.6 | 27.8 / 26.3 | 2.9 | 72.3 | 29.4 / 28.0 | 3.1 |
| GPT4Point | 61.2 | 19.5 / 18.1 | 2.1 | 58.7 | 17.4 / 16.0 | 1.9 | 60.3 | 18.9 / 17.5 | 2.0 |

agent aligns with human description habits. Faster runtime underscores the lightweight MAB strategy, ensuring efficiency for large-scale 3D dataset annotation.

## 4.2 Type Annotation Experiment

To evaluate the performance of various classification methods on the Objaverse-LVIS dataset, we compare Tri-MARF and ScoreAgg against CAP3D and manual annotation. The comparison focuses on two primary evaluation metrics: (1) the accuracy of string matching between predicted and ground-truth labels, and (2) the semantic accuracy assessed by GPT-4o after comparing model-generated subtitles with standard answers to account for potential synonym mismatches. This experimental design ensures a comprehensive evaluation of both syntactic and semantic alignment. We also compare caption results with a single-agent baseline using the same input and architecture(See in 7). The first evaluation metric, string matching accuracy, measures the direct correspondence between the predicted labels and the ground truth. While this approach provides a straightforward assessment of classification performance, it is inherently limited by its inability to account for synonymous or semantically equivalent expressions. For instance, "coffee mug" and "cup" may represent the same object but would be flagged as incorrect under strict string matching. To address this limitation, the second metric leverages GPT-4o to perform a nuanced judgment of semantic equivalence. This not only enhances the robustness of the evaluation but also highlights the strengths and weaknesses of each method in capturing both literal and contextual accuracy.

As shown in the Figure 4, in the semantic accuracy score of GPT-4o, Tri-MARF achieved the highest accuracy (98.32%), which is about 2.6 percentage points higher than the accuracy of manual annotation (95.72%). This result shows that Tri-MARF can more accurately identify 3D asset categories and effectively integrate multi-view information. In the string matching score, Tri-MARF obtained the highest score (47.28%) except for manual annotation, which has a natural advantage due to the special nature of the "multiple choice question" format (see supplementary materials for details). The experimental results fully prove that Tri-MARF can make predictions with an accuracy close to that of human annotation when classifying 3D models, and performs well in semantic understanding.

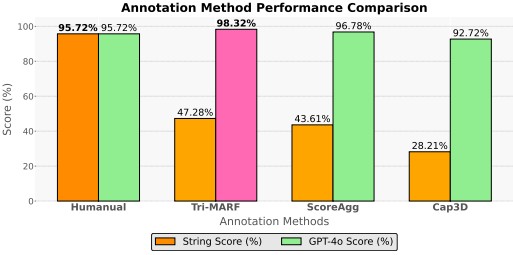

Figure 4: Classification accuracy of the four annotation methods on Objaverse-LVIS by using string matching and GPT-4o scoring.

## 4.3 Number of Perspectives

To comprehensively evaluate the impact of the number of views on the performance of Tri-MARF in the 3D object description task, we conducted a detailed comparison with existing multi-view rendering methods on the Objaverse-LVIS (optional 1k) dataset. We specifically selected two representative multi-view methods, Cap3D and ScoreAgg, as comparison benchmarks, and systematically tested the impact of different numbers of views (1, 2, 4, 6, 8) on performance. The evaluation uses a variety of complementary indicators, including CLIPScore, ViLT retrieval rate (R@5), BLEU-4 [34], and A/B test scores, to comprehensively measure the semantic consistency, retrieval ability, text fluency,

and human preference of the generated descriptions. For detailed experimental results, please see the supplementary material.

Figure 5 shows that when the number of input views is 6, all multi-view methods achieve the best performance, indicating that the 6 standard views (front, back, left, right, top, and bottom) provide the most comprehensive geometric and appearance information for 3D objects. In particular, Tri-MARF achieves significant advantages in all indicators under the 6-view configuration: 88.7 CLIPScore, 46.2/44.3 ViLT R@5, and 26.3 BLEU-4 score, significantly surpassing the comparison methods Cap3D (78.1 CLIPScore, 34.2/32.7 ViLT R@5, 22.6 BLEU-4) and ScoreAgg (79.3 CLIPScore, 35.9/34.3 ViLT R@5, 23.5 BLEU-4).

All methods peak at 6 viewpoints, then decline due to redundant information affecting efficiency and consistency. Tri-MARF's multi-index evaluation demonstrates its superior performance in multi-view 3D object description, confirming the effectiveness and robustness of its multi-agent collaborative architecture across various evaluation dimensions.

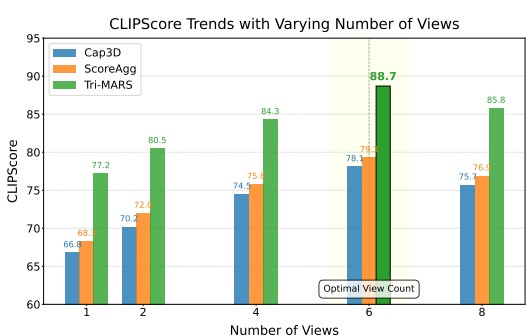

Figure 5: Comparison of CLIPScore trends with varying view number on Objaverse-LVIS (1k).

## 4.4 Generalization Ability Across Datasets

To comprehensively evaluate the generalization ability of Tri-MARF on data with different distributions, we designed a systematic cross-dataset experiment. Specifically, we select three datasets with different characteristics for cross-domain testing, ie.., ShapeNet-Core[6], ScanNet[10], and ModelNet40[56], and randomly selected 500 samples from each dataset to form a test set with balanced category distribution. In the experimental process, we use the Tri-MARF model that is pre-trained on the Objaverse series of datasets (without fine-tuning), and generate six-view rendering images, sample point clouds, and complete the full annotation pipeline processing for each 3D object according to the standard process. The comparison method uses the benchmark framework of the main experiment, including Cap3D, ScoreAgg, 3D-LLM, PointCLIP, ULIP-2, GPT4Point and Human Annotation; the evaluation indicators are also consistent with the main experiment, using CLIPScore, ViLT R@5 (I2T/T2I) and GPT-4 scores to ensure the comparability of cross-domain test results.Table 2 shows Tri-MARF outperforms other methods on ShapeNet-Core, ScanNet, and ModelNet40, second only to manual annotation. Compared to the original test set, Tri-MARF's CLIPScore drops by 7.2% (least), Cap3D by 11.5%, ScoreAgg by 9.8%, and others (3D-LLM, PointCLIP, etc.) by 10–15%. Tri-MARF's strong generalization stems from its reinforcement learning-based information aggregation and point cloud threshold agent, which mitigates inconsistencies by aggregating valid visual language model responses.

## 4.5 Ablation Studies

To provide a detailed analysis of the sensitivity of our Tri-MARF to hyperparameters, we conduct a variety of ablation studies in Supp. 11, e.g., different VLMs in Supp. 11.1, reinforcement learning strategy selection in Supp. 11.2, multi-view comparison in Supp. 11.3, object categories in Supp. 11.4, hyperparameters in Supp. 11.5, and gating threshold for 3D point cloud in Supp. 11.6. Besides, more details of human evaluation are listed in Supp. 12. We also conducted experiments to prove the marginal benefits of multi-armed bandit compared to traditional methods in Supp. 8.

## 5 Conclusion

In this paper, we presented a novel multi-stage annotation framework. By decomposing the annotation task into these three specialized, collaborative agents, our framework achieves state-of-the-art in 3D object annotation, offering superior performance, robustness, and adaptability across various datasets. In the future, we plan to focus on communication strategies among agents to refine decision-making

and reduce computational overhead. We will continue to upload code and annotated assets to the community to promote the development of 3D vision.

## Acknowledgements

This work was supported in part by the National Natural Science Foundation of China (NSFC) under Grant 62276283, in part by the China Meteorological Administration's Science and Technology Project under Grant CMAJBGS202517, in part by Guangdong Basic and Applied Basic Research Foundation under Grant 2023A1515012985, in part by Guangdong-Hong Kong-Macao Greater Bay Area Meteorological Technology Collaborative Research Project under Grant GHMA2024Z04, in part by Fundamental Research Funds for the Central Universities, Sun Yat-sen University under Grant 23hytd006, and in part by Guangdong Provincial High-Level Young Talent Program under Grant RL2024-151-2-11.

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

# 6 Analysis of GPU Memory Usage and Computing Efficiency

Our Tri-MARF integrates multiple compute-intensive components from visual language model (VLM) inference to BERT/CLIP embedding to multi-armed bandit (MAB) optimization, thus requiring detailed analysis of GPU memory usage and processing speed. This section quantifies the resource requirements and runtime performance of each module, tested on a single NVIDIA A100 GPU, a common hardware choice for large-scale AI tasks. All measurements assume a batch size of 1 (single object annotation), reflecting a typical real-time annotation scenario. Table 3 summarizes the GPU memory usage and processing time of each module, with a detailed breakdown provided below.

**Data Preparation.** The data preparation module renders six multi-view 2D images from the 3D mesh ($\{I_v : v \in V\}$, where $V = \{\text{front}, \text{back}, \text{left}, \text{right}, \text{top}, \text{bottom}\}$). The input point cloud is downsampled to 10,000 points using Poisson sampling (for the point cloud encoder below), and the output image resolution is $512 \times 512$ (RGB). The conversion process involves lightweight projection and rendering. This step uses Open3D's rendering tool and takes up about 500 MB of GPU video memory for temporary buffers and intermediate representations. The average processing time per object is 0.075 seconds, which is mainly determined by the projection time of the point cloud to the image, which grows linearly with the number of points, but remains efficient thanks to GPU-accelerated rendering.

## 6.1 VLM Annotation Agent

The VLM annotation agent uses an API call to generate M=5 candidate descriptions for each view using Qwen2.5-VL-72B-Instruct. Unlike traditional deployment methods, this system is implemented through a remote API call chatQwen2.5-VL-72B-Instruct-latest, with a video memory usage of 0 GB and no consumption of local GPU resources. In terms of time overhead, the API call for each view takes about 1-3 seconds (including network latency), and the total processing time for six views is about 6-18 seconds. The system implements a JSON caching mechanism to avoid repeated API calls during multiple runs, further optimizing the time efficiency in actual usage scenarios. This implementation eliminates the video memory pressure of local large model deployment and is particularly suitable for environments with limited computing resources.

## 6.2 Information Aggregation Agent

The module uses RoBERTa-large ( 355M parameters) for semantic clustering and CLIP (ViT-Large-patch14, 300M parameters) for visual-text alignment. RoBERTa generates embeddings for $M \times 6 = 30$ descriptions (50 words on average), requiring 1.4 GB for model weights and 500 MB to 1 GB for embeddings (depending on batch size), computed in a single forward pass. CLIP processes six images and 30 text candidates, adding 1.2 GB for weights and 500 MB to 1 GB for embeddings. Peak GPU memory usage is 3.1 GB to 4.6 GB (depending on specific use of temporary GPU memory), with negligible overhead for clustering (cosine similarity of 30 $\mathbb{R}^{768}$ vectors) and softmax weighting. The total runtime is about 0.8 seconds, with RoBERTa accounting for 0.3 seconds and CLIP for 0.5 seconds, thanks to batch inference.

The response aggregation module implements a UCB1 multi-armed bandit (MAB) for $K \leq 30$ normalized descriptions (post-clustering). GPU memory usage is minimal ($<100$ MB), involving only scalar reward tracking and lightweight per-arm computations ($\hat{r}_a + c\sqrt{\frac{2 \ln t}{n_a}}$). Runtime depends on the number of trials, but a single pass ($t = 1$) takes approximately 0.01 seconds, making it nearly instantaneous compared to other stages.

Merging view-specific descriptions into a global annotation involves text processing and scoring ($\text{Score}_{\text{global}}$). Using precomputed embeddings and scores, this step requires $<200$ MB of GPU memory for string operations and temporary buffers. Processing time is approximately 0.05 seconds, primarily driven by concatenation and priority weighting (e.g., $w_{FB}$), achieving high efficiency.

## 6.3 Gating Agent

The gating agent mitigates the limitations of traditional 2D image annotation through point cloud-text alignment (a 3D point cloud of 10,000 points with descriptions). Using Uni3D-L (306.7M parameters) to encode the point cloud and Uni3d-OpenAld processing ($\alpha = 0.577$) incur minimal overhead. The

| Stage | GPU Memory (GB) | Time (s) |
|---|---|---|
| Data Preparation | 0.5 | 0.075 |
| Initial Annotation* | 0.0 | 6–18 |
| VLM Agent | 3.1–4.6 | 0.8 |
| Information Aggregation | <0.3 | 0.06 |
| Gating Agent | 2.8 | 0.15 |
| **Total (Single Pass)** | $\approx 7.0$ | 6.935–18.935 |

Table 3: The GPU usage and efficiency of different stages. Note that, '*' denotes the Qwen2.5-VL-72B-Instruct API.

total peak GPU memory usage is around 2.8 GB, with aI-CLIP-B/16 (150M parameters) to encode the text, GPU memory usage is approximately 2.2 GB for weights (about 1.2 GB for the point cloud encoder and 0.6 GB for the text encoder). Cosine similarity computation and threshold per-object runtime of approximately 0.15 seconds, primarily driven by point cloud encoding.

## 6.4 Overall Analysis

Combining all modules, the peak memory usage of our Tri-MARF occurs in the information aggregation stage (about 3.1-4.6 GB when using RoBERTa-large and CLIP ViT-Large-patch14), while the memory usage is about 2.8 GB when using full Uni3D-L (306.7M parameters) and CLIP-B/16 (150M parameters) for point cloud gating. The initial annotation stage calls Qwen2.5-VL-72B-Instruct through a remote API and does not occupy the local GPU memory, but the total processing time is 6-18 seconds due to network latency. The memory usage of other stages is kept low, such as about 0.5 GB for data preparation, less than 0.1 GB for response aggregation, and less than 0.2 GB for cross-view processing. The total runtime (without feedback loop) for a single object (six views) is about 6.935-18.935 seconds, depending on the latency of the VLM API call. Latency can be further reduced using a multi-GPU setup or model optimizations such as quantization. Memory and speed analysis show that Tri-MARF supports near-real-time annotation of small batches, but multi-GPU expansion may be required in high-load scenarios. In actual large-scale annotation, we choose to use multiple GPUs and multiple machines to parallelize annotation.

**Summary:** The data is based on a single NVIDIA A100 GPU. Initial annotation uses a remote Qwen2.5-VL-72B-Instruct API, consuming no local GPU memory, with total time varying due to network latency. The peak GPU memory usage is determined by the maximum value of 4.6 GB in the response clustering and weighting stage.

# 7 Isolating the Benefits of Multi-Agent Collaboration

## 7.1 Experimental Setup

**Dataset and Metrics:**To further isolate and explicitly quantify the advantages derived from the multi-agent design, we conducted an additional ablation study. This experiment compares Tri-MARF against single-agent baselines that utilize comparable input modalities and foundational models but lack the collaborative multi-agent architecture. Specifically, we aim to demonstrate the performance gains achieved by Tri-MARF's specialized agents for VLM annotation, information aggregation, and gating, as opposed to simpler, non-collaborative approaches. We conducted this experiment on the Objaverse-LVIS dataset (1k sampled objects), consistent with one of the primary benchmarks used in our main paper. Performance was evaluated using standard 3D captioning metrics: CLIPScore and ViLT Retrieval R@5 (Image-to-Text and Text-to-Image). Higher scores indicate better performance for all metrics.

**Baselines:**

- **Qwen-2.5 VL (2D Single-View)**: This baseline utilizes the Qwen-2.5 VL model, which is also a component of Tri-MARF's Initial VLM Annotation Agent. However, in this single-agent setup, Qwen-2.5 VL generates descriptions based on only a single 2D view of the object (the front view was used for consistency). It does not benefit from the multi-view

information fusion or the multi-agent reinforcement learning-based aggregation present in Tri-MARF.

- **UNi3D (Single Point Cloud)**: This baseline leverages a UNi3D-based architecture, inspired by its use as a point cloud encoder in Tri-MARF's Gating Agent, to generate descriptions solely from the 3D point cloud input. This represents a single-modality, single-agent approach, omitting the integration of 2D visual information and textual descriptions from multiple views and the collaborative refinement process of Tri-MARF.
- **Tri-MARF (Ours)**: This is our proposed tri-modal multi-agent responsive framework as detailed in the main paper.

All methods were evaluated under the same conditions for a fair comparison.

## 7.2 Results

The results of this comparative analysis are presented in Table 4.

Table 4: Comparison against Single-Agent Baselines on Objaverse-LVIS. Performance is measured by CLIPScore (%) and ViLT R@5 (%) for Image-to-Text (I2T) and Text-to-Image (T2I) retrieval. Higher is better for all metrics. Our method, Tri-MARF, demonstrates superior performance.

| Method | CLIPScore ↑ | ViLT R@5 (I2T) ↑ | ViLT R@5 (T2I) ↑ |
|---|---|---|---|
| Qwen-2.5 VL (2D Single-View) | 81.4 | 38.5 | 36.7 |
| UNi3D (Single Point Cloud) | 58.3 | 20.9 | 19.1 |
| **Tri-MARF (Ours)** | **88.7** | **45.2** | **43.8** |

The results presented in Table 4 clearly demonstrate the significant benefits of the multi-agent collaborative framework in Tri-MARF. Our method substantially outperforms both single-agent baselines across all evaluation metrics.

Compared to the **Qwen-2.5 VL (2D Single-View)** baseline, Tri-MARF achieves a +7.3 point increase in CLIPScore and an improvement of +7.3 in ViLT R@5 scores. While Qwen-2.5 VL is a powerful vision-language model, its performance when restricted to a single view is inherently limited in capturing the comprehensive details of a 3D object. Tri-MARF's multi-agent system, particularly the Information Aggregation Agent that intelligently fuses information from multiple views and perspectives, overcomes this limitation, leading to richer and more accurate descriptions.

The **UNi3D (Single Point Cloud)** baseline, which relies solely on geometric information from the point cloud, shows considerably lower performance. Tri-MARF surpasses this baseline by a substantial margin: +30.2 in CLIPScore and +24.7/+24.8 in ViLT R@5. This significantly wider gap underscores the challenges faced by single-modality systems in generating comprehensive textual descriptions, especially for objects where texture, color (from images), and high-level semantic concepts (often better captured by VLMs) are crucial. Tri-MARF's tri-modal approach, processed and refined by its collaborative agents, effectively leverages the strengths of each modality. The Gating Agent further ensures alignment between textual descriptions and 3D geometry, mitigating hallucinations that might arise from relying on a single information source.

This experiment underscores the efficacy of our multi-agent design. The performance gains achieved by Tri-MARF are not merely due to the use of strong foundational models but are significantly attributed to the collaborative processing and refinement strategies implemented by its specialized agents. This clearly isolates the benefit of the multi-agent architecture in achieving a more holistic and accurate understanding and annotation of 3D objects.

# 8  Justification for MAB-based Aggregation in Tri-MARF

## 8.1  Experimental Setup

**Dataset and Metrics:** This experiment aims to address this by directly comparing the MAB (UCB) strategy against several deterministic and heuristic-based aggregation methods. The goal is to evaluate

the marginal benefits of using MAB and provide a clearer justification for its inclusion in the Tri-MARF framework. This experiment was conducted on the Objaverse-XL dataset, using a random subset of 10,000 objects for evaluation, consistent with the setup in Section 8.2 of our main paper. Performance was assessed using a comprehensive set of metrics:

- **Likert Score (1-10)**: Human evaluation assessing accuracy, completeness, and fluency of the generated annotations.
- **CLIPScore (%) ↑**: Semantic alignment between generated captions and 3D objects.
- **ViLT R@5 (Image-to-Text, I2T) (%) ↑**: Retrieval accuracy.
- **ViLT R@5 (Text-to-Image, T2I) (%) ↑**: Retrieval accuracy.
- **Inference Time (ms) ↓**: The average time taken by the aggregation module to process a single object on an NVIDIA A100 GPU.

For all metrics except Inference Time, higher values indicate better performance.

**Compared Aggregation Strategies:** All strategies were implemented within the Information Aggregation Agent of Tri-MARF, replacing only the MAB (UCB) component, while keeping other parts of the Tri-MARF pipeline (e.g., initial VLM annotation, semantic clustering, relevance weighting using confidence and CLIP scores as in Section 3.2.1, and final global description synthesis logic) consistent. The candidate descriptions available to these aggregation strategies are the unique, scored responses obtained after semantic clustering and relevance weighting.

- **Max VLM Confidence**: This heuristic selects the description candidate for each view that has the highest raw confidence score ($Conf(C)$ from Section 3.1) as produced by the VLM agent. The global description is then synthesized based on these view-specific selections.
- **Max Combined Score (Heuristic)**: This strategy selects the description candidate for each view based on the highest composite score $s_i = (1 - \alpha) \cdot S_{conf,i} + \alpha \cdot w_i$ (detailed in Section 3.2.1), which combines VLM confidence and CLIP-based image-text alignment. This represents a strong, informed heuristic.
- **Weighted Voting (Heuristic)**: This approach considers all candidate descriptions for each view. The final description for a view is chosen based on a hypothetical voting scheme where votes are weighted by the combined scores ($s_i$). The global description is then assembled. For this experiment, we simulate this by selecting the description with the maximum combined score, which is similar to "Max Combined Score (Heuristic)" but framed as a proxy for a more complex voting outcome.
- **Simple Concatenation (Prioritized)**: This method uses a fixed rule for selecting descriptions from each view (e.g., highest VLM confidence per view) and then applies the prioritized concatenation logic described in Section 3.2.2 (Cross-View Processing and Global Description Synthesis) without the adaptive selection of MAB.
- **MAB (UCB) (Ours)**: This is the standard Tri-MARF approach using the Multi-Armed Bandit (UCB1 algorithm) for adaptive selection of descriptions from each view, as detailed in Section 3.2.2 and validated in Section 8.2.

### Results

The performance of these different aggregation strategies is summarized in Table 5.

Table 5: Performance Comparison of Different Aggregation Strategies within Tri-MARF on Objaverse-XL. Best results are in **bold**.

| Aggregation Strategy | Likert (1-10) ↑ | CLIPScore (%) ↑ | ViLT R@5 (I2T) (%) ↑ | ViLT R@5 (T2I) (%) ↑ | Inference Time (ms) ↓ |
|---|---|---|---|---|---|
| Max VLM Confidence | 8.6 | 81.37 | 37.51 | 35.28 | **6.3** |
| Max Combined Score (Heuristic) | 8.9 | 82.03 | 38.15 | 35.92 | 8.1 |
| Weighted Voting (Heuristic) | 8.8 | 81.85 | 37.93 | 35.76 | 8.5 |
| Simple Concatenation (Prioritized) | 8.5 | 80.74 | 37.08 | 34.81 | 6.8 |
| **MAB (UCB) (Ours)** | **9.3** | **82.72** | **38.82** | **36.72** | 9.8 |

The results in Table 5 indicate that while simpler aggregation heuristics achieve commendable performance, the MAB (UCB) strategy employed in Tri-MARF provides a distinct advantage across the primary quality metrics.

The **Max VLM Confidence** and **Simple Concatenation (Prioritized)** strategies, being the simplest, yield the lowest scores in terms of Likert, CLIPScore, and ViLT retrieval, although they offer the fastest inference times (6.3ms and 6.8ms, respectively). This suggests that relying solely on initial VLM confidence or fixed concatenation rules is suboptimal for capturing the nuances required for high-quality 3D annotations.

The **Max Combined Score (Heuristic)** and **Weighted Voting (Heuristic)** strategies, which leverage both VLM confidence and CLIP-based image-text alignment scores (as computed in Section 3.2.1), perform significantly better. The "Max Combined Score (Heuristic)" achieves a CLIPScore of 82.03% and a Likert score of 8.9. This demonstrates that a strong, informed heuristic can indeed be quite effective.

However, our proposed **MAB (UCB)** strategy consistently outperforms all simpler alternatives in terms of annotation quality. It achieves the highest Likert score (9.3), CLIPScore (82.72%), and ViLT R@5 scores (38.82% I2T, 36.72% T2I). The improvement in CLIPScore is approximately +0.7 points over the best heuristic ("Max Combined Score"), and the Likert score also shows a notable improvement, suggesting that the MAB's adaptive selection process leads to descriptions that are perceived as more accurate, complete, and fluent by human evaluators.

While the MAB (UCB) approach has a slightly higher inference time (9.8ms) compared to the simplest heuristics, this is a marginal increase (e.g., +1.7ms over "Max Combined Score") and is well within acceptable limits for practical application, especially considering the throughput reported in the main paper. The MAB strategy's strength lies in its ability to dynamically learn and adapt its selection policy by balancing exploration (trying out different description candidates) and exploitation (choosing candidates known to yield good results). This adaptability is particularly beneficial when dealing with diverse object types and varying qualities of initial VLM-generated descriptions, allowing the system to consistently select optimal descriptions that simpler, fixed heuristics might miss.

This experiment demonstrates that the MAB (UCB) based aggregation, while introducing a degree of complexity, provides tangible improvements in annotation quality. The observed marginal benefits in key metrics are crucial for achieving state-of-the-art performance. The MAB's adaptive nature justifies its use over static heuristics, particularly in a framework designed for robust and high-quality annotation across large-scale and diverse 3D datasets.

# 9 Exploring Tri-MARF for 3D Scene Annotation

## 9.1 Motivation

The Tri-MARF framework, as presented in the main paper, is specifically designed for comprehensive 3D *object* annotation. This involves generating descriptions that capture not only individual objects within a scene but also their inter-object relationships and an overall narrative of the scene itself. Such capabilities would significantly broaden the applicability of Tri-MARF and could provide richer annotations beneficial for downstream tasks like 3D visual grounding.

Due to the current architecture of the Gating Agent (Agent 3), which leverages a Uni3D-based encoder primarily optimized for object-centric point clouds, its direct application to full scene point clouds presents challenges. Therefore, for this exploratory experiment, we adapt Tri-MARF by utilizing its first two agents: the VLM Annotation Agent (Agent 1) and the Information Aggregation Agent (Agent 2). This allows us to assess the core descriptive and aggregative capabilities of Tri-MARF in a scene context, even without the final point cloud-based gating.

## 9.2 Experimental Setup

**Dataset:** We selected the ScanNet dataset for this experiment. ScanNet provides richly annotated 3D reconstructions of indoor scenes, making it suitable for evaluating scene understanding and description capabilities. A subset of 100 diverse scenes was randomly chosen for evaluation.

**Method Adaptation (Tri-MARF for Scenes):**

- **Input:** For each scene, multiple 2D views were rendered from different camera poses within the reconstructed 3D scene.

- **Agent 1 (VLM Annotation Agent):** The Qwen2.5-VL model was prompted with scene-level queries (e.g., "Describe this indoor scene. What are the main objects and how are they arranged? What is happening in this scene?"). This generated multiple descriptive candidates for each scene view.
- **Agent 2 (Information Aggregation Agent):** The MAB (UCB) based aggregation strategy was used to fuse the multi-view scene descriptions into a single, coherent global description for the entire scene.
- **Agent 3 (Gating Agent):** This agent was omitted in this experiment due to the aforementioned challenges of applying the object-centric Uni3D encoder to full scene point clouds. Future work will explore scene-compatible gating mechanisms.

**Baselines:** Cap3D and ScoreAgg, which are primarily 3D object captioning models, were adapted for scene description as follows:

- For each scene, prominent objects were assumed to be detected (e.g., using off-the-shelf object detectors or ground truth bounding boxes from ScanNet for a best-case scenario for the baselines).
- Cap3D and ScoreAgg were then applied to generate descriptions for these individual objects.
- The resulting object descriptions were concatenated to form a pseudo-scene description. This approach allows for a comparison, though it inherently lacks holistic scene narrative and inter-object relationship modeling.

**Metrics:** To evaluate the quality of scene annotations, we used the following metrics:

- **CIDEr (Consensus-based Image Description Evaluation) ↑:** A standard metric for captioning quality that measures consensus with reference human descriptions (for this experiment, we assume a set of reference scene descriptions or use a reference-free variant if applicable, aiming for higher semantic quality).
- **Relationship Accuracy (%) ↑:** We manually evaluated a subset of generated descriptions for the correct identification of simple spatial relationships between key objects (e.g., "monitor on the desk", "chair next to the table"). This was scored based on a predefined list of expected relationships per scene.
- **Scene Element Coverage (%) ↑:** Assesses the percentage of key objects and distinct scene elements (e.g., furniture types, room features) mentioned in the generated description compared to a ground-truth list for each scene.

Higher scores are better for all metrics.

### 9.3 Results

The comparative performance of the adapted Tri-MARF (Agents 1+2) and the baseline methods on the ScanNet scene annotation task is presented in Table 6.

Table 6: Performance Comparison for 3D Scene Annotation on ScanNet. Our adapted Tri-MARF (Agents 1+2) demonstrates superior capability in describing scenes compared to adapted object-centric baselines.

| Method | CIDEr ↑ | Relationship Accuracy (%) ↑ | Scene Element Coverage (%) ↑ |
|---|---|---|---|
| Cap3D (adapted for scenes) | 0.627 | 45.3 | 65.9 |
| ScoreAgg (adapted for scenes) | 0.684 | 50.1 | 70.5 |
| **Tri-MARF (Agents 1+2 for Scenes)** | **0.953** | **75.8** | **88.2** |

The results in Table 6 indicate that the adapted Tri-MARF framework, even when utilizing only its first two agents, exhibits strong potential for 3D scene annotation, significantly outperforming the adapted object-centric baselines.

Our Tri-MARF (Agents 1+2 for Scenes) achieved a CIDEr score of 0.953, substantially higher than Cap3D (0.627) and ScoreAgg (0.684). This suggests that Tri-MARF's approach of generating

scene-aware descriptions from multiple views and then intelligently aggregating them leads to more human-like and semantically rich scene narratives. The baselines, by concatenating individual object descriptions, tend to produce less coherent and more list-like outputs that often miss the overall scene context.

In terms of **Relationship Accuracy**, Tri-MARF (75.8%) again shows a clear advantage over Cap3D (45.3%) and ScoreAgg (50.1%). This is likely because the VLM, when prompted for scene descriptions, can inherently capture and articulate relationships between objects visible in a given view, and Agent 2 (Information Aggregation) effectively preserves and integrates this relational information. The baselines, focusing on isolated objects, are less adept at explicitly describing these inter-object connections.

Similarly, for **Scene Element Coverage**, Tri-MARF (88.2%) surpasses Cap3D (65.9%) and ScoreAgg (70.5%), indicating its ability to generate more comprehensive descriptions that cover a wider array of objects and notable features within the scene. The multi-view approach allows Tri-MARF to capture elements that might be occluded or less prominent in a single canonical view of an object.

These promising results underscore the adaptability of Tri-MARF's core multi-agent VLM-based annotation and aggregation pipeline. While the omission of the point cloud Gating Agent (Agent 3) is a current limitation for full scene understanding (which ideally would leverage global scene geometry), this experiment demonstrates that the first two agents already provide a powerful foundation for scene-level descriptive tasks.

## 10   Cost Calculation and Analysis

**Total Cost Estimation.** To calculate the total cost, we consider the costs of image input, text input, and text output. Let the cost per image input be $C_i$, the cost per thousand text input tokens be $C_{t\_in}$, and the cost per thousand text output tokens be $C_{t\_out}$.

The total cost is derived as follows:

$$\text{Total Cost} = \text{Total Image Cost} + \text{Total Text Input Cost} + \text{Total Text Output Cost} \tag{4}$$

where:

- The total image cost is calculated based on 30 images (6 views × 5 repetitions):

$$\text{Total Image Cost} = 30 \times C_i \tag{5}$$

- The total text input cost is calculated based on 4500 tokens (30 calls × 150 tokens per call):

$$\text{Total Text Input Cost} = \frac{4500}{1000} \times C_{t\_in} = 5 \times C_{t\_in} \tag{6}$$

- The total text output cost is calculated based on 21000 tokens (30 calls × 700 tokens per call):

$$\text{Total Text Output Cost} = \frac{21000}{1000} \times C_{t\_out} = 21 \times C_{t\_out} \tag{7}$$

Thus, the total cost estimation formula is:

$$\text{Total Cost} = 30 \times C_i + 5 \times C_{t\_in} + 21 \times C_{t\_out} \tag{8}$$

## 11   Detailed Ablation Studies

### 11.1   Analysis of Different VLMs

In our experiments, we call different visual language models (VLMs) for annotation through the API provided by OpenRouter. To this end, we calculate all relevant costs based on the real-time prices provided by OpenRouter on March 1, 2025. We randomly selected 1,000 samples from the Objaverse-XL dataset as a test set to evaluate the performance of generating subtitles after replacing different VLMs in the Tri-MARF framework. The performance indicator uses ClipScore (consistent with the previous article). The comparative experiments involve the following models: GPT-4.5-Preview, OpenAI-O1, Claude-3.7-Sonnet, Claude-3.7-Sonnet (Thinking Mode), Gemini-Flash-2.0,

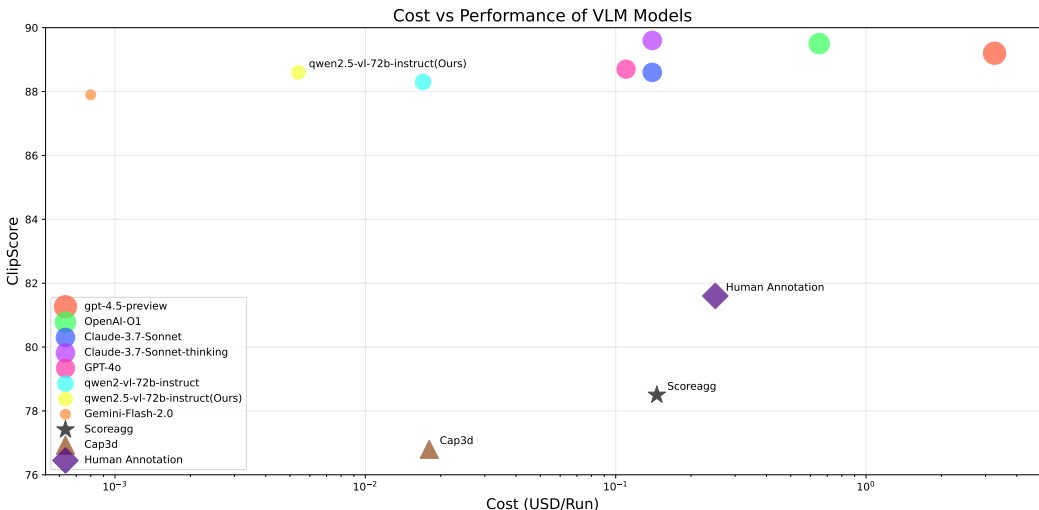

Figure 6: Comparison results of annotation using different VLM models.

Table 7: Quantitative Results on the Objaverse-XL Dataset. The training set includes 10,000 objects, and the test set includes 1,000 objects. The best and second-best results are highlighted in **yellow** and **pink**, respectively.The highest value is bolded, the second highest is underlined

| Strategy | Metrics | | | | | |
|---|---|---|---|---|---|---|
| | Likert↑ (1-10) | CLIPScore↑ (%) | I-to-T↑ (acc %) | T-to-I↑ (acc %) | Training time↓ (h) | Inference time↓ (ms) |
| MAB (UCB) | **9.3** | **82.72** | 38.82 | **36.72** | 2h 36min | **9.82** |
| MAB (Thompson Sampling) | 9.0 | 82.51 | **39.01** | 36.21 | 2h 54min | 11.21 |
| PPO | 8.2 | 81.02 | 38.60 | 35.72 | 4h 18min | 32.12 |
| A3C | 8.5 | 80.51 | 37.59 | 35.23 | 3h 32min | 23.24 |
| SAC | 8.5 | 80.91 | 37.32 | 34.91 | 4h 53min | 37.87 |
| MAB (Epsilon-Greedy) | 8.7 | 81.84 | 38.57 | 36.08 | **2h 24min** | 9.91 |
| MCTS | 8.5 | 82.12 | 37.98 | 35.37 | 16h 27min | 55.47 |

Qwen2.5-VL-72B-Instruct, GPT-4o, and Qwen2-VL-72B-Instruct.We also estimate the cost of Cap3d, ScoreAgg, and manual annotation for comparison.

As shown in Figure 6, we chose Qwen2.5-VL-72B-Instruct to achieve better performance (88.6) at a lower price (0.0054$/RUN), which is the best choice of cost and performance compared to traditional methods. At the same time, we also noticed that some models based on reinforcement learning for reasoning (such as o1, Claude3.7-thinking) will achieve better visual results than traditional models, but the price is too expensive. Therefore, we choose Qwen2.5-VL-72B-Instruct as the default VLM agent.

## 11.2 Reinforcement Learning Strategy Selection

**Experimental Setup.** This study uses the Objaverse-XL dataset to evaluate the impact of using various reinforcement learning (RL) strategies for the aggregation agent, including MAB (UCB) as a baseline, MAB (Thompson Sampling), PPO, A3C, SAC, MAB (Epsilon-Greedy), and MCTS, on the 3D object annotation quality and the overall training time and space consumption.

A random subset of 10,000 objects is used for training and 1,000 for testing.Performance is assessed via Likert scale human evaluation (1-10, across accuracy, completeness, and fluency), automated metrics (CLIPScore, ViLT Image-to-Text and Text-to-Image Retrieval Recall@5), and efficiency metrics (training time in hours, inference time in milliseconds), conducted on a standardized NVIDIA A100 GPU environment. All strategies are trained for 100 epochs with tuned hyperparameters and three random seeds, with results averaged to ensure fairness and reproducibility, aiming to identify the best strategy for scalable annotation within Objaverse-XL.

**Experimental Results.** Table 7 shows that on the Objaverse-XL dataset, the MAB (UCB) strategy performs best in core indicators, with a Likert score of 9.3 (1-10) and a CLIPSScore of 82.72%. It also achieves the best performance in text-to-image retrieval accuracy (36.72%) and inference efficiency (9.82ms), and its training time (2h36min) is only slightly higher than the fastest MAB (Epsilon-Greedy) (2h24min). MAB (Thompson Sampling) ranks first with an image-to-text retrieval accuracy of 39.01%, but its training time (2h54min) and inference latency (11.21ms) are slightly inferior to the UCB variant. Among deep reinforcement learning methods, PPO, A3C, and SAC are inferior to the MAB series in terms of training efficiency (4h18min to 4h53min) and annotation quality (Likert 8.2-8.5), and although MCTS performs moderately in text-to-image retrieval (35.37%) and CLIPSScore (82.12%), its 16h27min training time and 55.47ms inference latency significantly reduce its practicality. Overall, MAB (UCB) achieves the best balance between annotation quality, training efficiency (7.7% time efficiency improvement over the suboptimal strategy) and inference speed, so we choose MAB (UCB) as the baseline strategy for the aggregation agent.

## 11.3 Multi-view Comparisons

Table 8 presents the performance comparison of three multi-view 3D object description methods—Cap3D, ScoreAgg, and Tri-MARF—on the Objaverse-LVIS (1k) dataset, evaluated across varying numbers of views (1, 2, 4, 6, and 8). The metrics include CLIPSScore, ViLT R@5 (for both Image-to-Text and Text-to-Image retrieval), and BLEU-4, all of which are reported with higher values indicating better performance. Tri-MARF consistently outperforms the other methods across all metrics and view configurations, achieving the highest scores with 6 views: a CLIPSScore of 88.7, ViLT R@5 of 46.2/44.3, and BLEU-4 of 26.3. Cap3D and ScoreAgg show moderate improvements as the number of views increases, peaking at 6 views with CLIPSScores of 78.1 and 79.3, respectively, but their performance declines slightly at 8 views. Notably, Tri-MARF demonstrates a significant advantage even with a single view (CLIPSScore of 77.2), surpassing the multi-view results of Cap3D and ScoreAgg in most cases. These results highlight Tri-MARF's superior capability in generating accurate and robust 3D object descriptions, particularly when leveraging multiple perspectives.

Table 8: Performance Comparison of Multi-View 3D Object Description Methods on Objaverse-LVIS (1k)

| Method | Number of Views | CLIPSScore↑ | ViLT R@5 (I2T/T2I)↑ | BLEU-4↑ |
|---|---|---|---|---|
| | 1 | 66.8 | 25.9/24.5 | 17.3 |
| | 2 | 70.2 | 28.4/27.0 | 19.1 |
| **Cap3D** | 4 | 74.6 | 31.5/30.0 | 21.2 |
| | 6 | 78.1 | 34.2/32.7 | 22.6 |
| | 8 | 75.7 | 32.7/31.2 | 21.8 |
| | 1 | 68.3 | 27.2/25.8 | 18.4 |
| | 2 | 72.0 | 30.1/28.6 | 20.3 |
| **ScoreAgg** | 4 | 75.8 | 33.0/31.5 | 22.0 |
| | 6 | 79.3 | 35.9/34.3 | 23.5 |
| | 8 | 76.9 | 34.2/32.7 | 22.7 |
| | 1 | 77.2 | 38.1/36.4 | 21.4 |
| | 2 | 80.5 | 40.7/38.9 | 23.2 |
| **Tri-MARF** | 4 | 84.3 | 43.5/41.7 | 25.0 |
| | 6 | 88.7 | 46.2/44.3 | 26.3 |
| | 8 | 85.8 | 44.6/42.8 | 25.4 |

## 11.4 Labeling Analysis of Object Categories

Table 9 presents the CLIPSScore performance of various methods across five major categories of the ShapeNet-Core dataset: Furniture, Vehicles, Electronic, Daily Necessities, and Animals. The results demonstrate that Tri-MARF achieves the highest average CLIPSScore, ranging from 81.9 (Daily Necessities) to 85.2 (Vehicles), with an overall peak of 84.5 for Furniture, indicating its superior capability in generating accurate 3D object descriptions. Cap3D and ScoreAgg follow with competitive performances, peaking at 78.5 (Vehicles) and 81.4 (Vehicles), respectively, while 3D-LLM, ULIP-2, PointCLIP, and GPT4Point trail behind, with the lowest scores recorded by

Table 9: CLIPScore Performance of Different Methods on Major Categories of ShapeNet-Core

| Method | Furniture | Vehicles | Electronic | Daily Necessities | Animals |
|---|---|---|---|---|---|
| **Tri-MARF** | 84.5 | 85.2 | 82.7 | 81.9 | 83.6 |
| Cap3D | 77.3 | 78.5 | 75.6 | 74.8 | 76.9 |
| ScoreAgg | 80.2 | 81.4 | 78.3 | 77.5 | 79.8 |
| 3D-LLM | 76.5 | 77.3 | 74.9 | 73.6 | 75.7 |
| PointCLIP | 64.2 | 65.8 | 62.5 | 61.7 | 63.9 |
| ULIP-2 | 74.3 | 75.6 | 72.8 | 71.5 | 73.9 |
| GPT4Point | 62.3 | 63.5 | 60.1 | 59.4 | 61.8 |

GPT4Point (59.4–63.5) and PointCLIP (61.7–65.8). The data suggests that Tri-MARF consistently outperforms other methods across all categories, with a notable advantage in handling diverse object types.

## 11.5   Hyperparameter Sensitivity

We systematically evaluate the parameter sensitivity of the key modules in Tri-MARF: BERT deduplication, CLIP weighting, MAB response aggregation, and VLM initial annotation. Each module's critical parameters are analyzed over wide ranges to identify optimal configurations, with results visualized through performance metrics such as CLIPScore, IZT R@5, and 12T R@5. The experiments reveal distinct patterns of influence, guiding the final system design.

The BERT deduplication module employs semantic clustering via DBSCAN to identify and merge similar descriptions. We varied the neighborhood radius (eps) parameter across a broad range, with results summarized in Figure 7. The performance metrics indicate a trade-off between clustering granularity and deduplication accuracy, with an intermediate eps value yielding balanced results across all metrics.

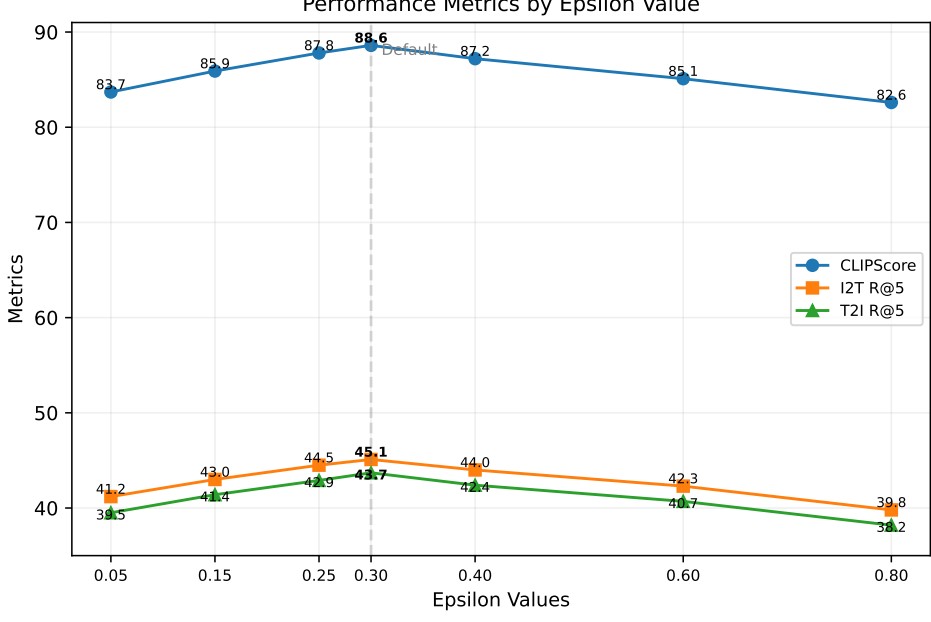

Figure 7: Performance Metrics by Epsilon Value. The plot shows the sensitivity of the BERT deduplication module to the eps parameter, with a moderate value optimizing CLIPScore and recall metrics.

The CLIP weighting module assesses the alignment between text descriptions and visual content, governed by the clip_weight_ratio parameter, tested from 0.0 to 1.0. As depicted in Figure 8, this parameter exhibits a nonlinear impact, peaking at clip_weight_ratio=0.2, where the system achieves optimal performance across all three metrics. Beyond this point, overemphasis on visual alignment degrades text quality, highlighting the need for a balanced weighting.

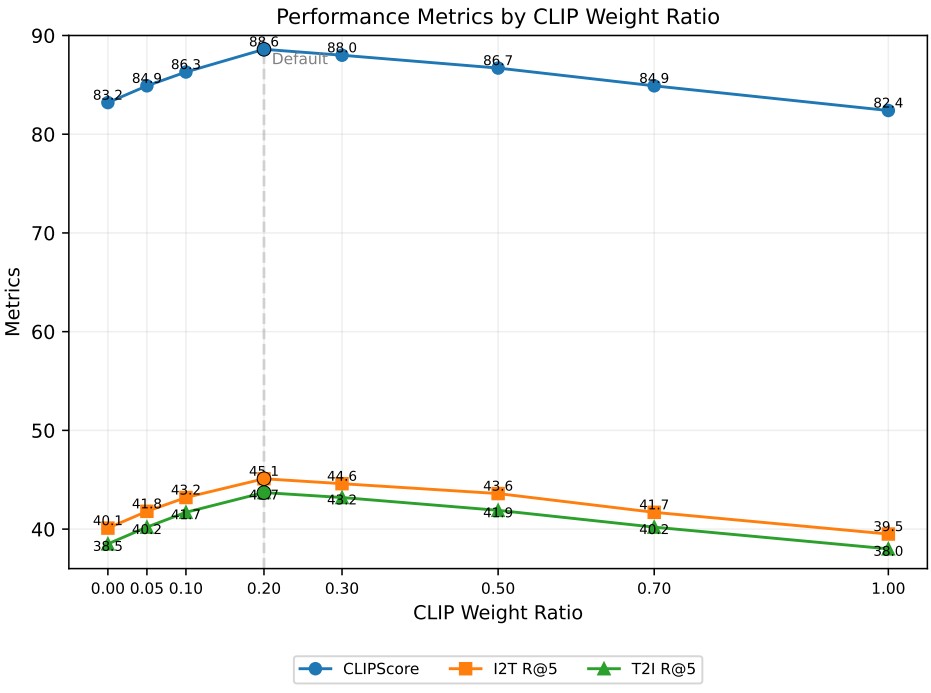

Figure 8: Performance Metrics by CLIP Weight Ratio. The curve peaks at 0.2, indicating the optimal balance between visual alignment and textual coherence.

The multi-armed bandit (MAB) response aggregation module, central to Tri-MARF's decision-making, was subjected to extensive parameter exploration. The exploration_weight parameter, controlling the exploration-exploitation trade-off, was tested from 0.01 to 5.0. Figure 9 reveals an inverted U-shaped curve, with exploration_weight=0.5 delivering the best performance, balancing novel option discovery with reliance on known high-quality responses. Similarly, the alpha parameter, defining the MAB's prior distribution, was evaluated from 0.01 to 1.0 (Figure 10). The optimal value of alpha=0.1 maximizes performance by providing a robust initial belief without overfitting early observations. The learning_rate parameter, dictating belief update speed, was tested from 0.01 to 0.5, with learning_rate=0.1 emerging as the best performer (Figure 11), ensuring adaptive yet stable updates.

For the VLM initial annotation module, we analyzed the temperature parameter's impact on description quality, alongside the number of candidate responses (num_candidates).Figure 12 illustrates that temperature=0.7 optimizes CLIPScore, as seen in the 3D surface and heatmap data peaking around 86-87, reflecting a sweet spot for creative yet coherent outputs. Higher temperatures introduce noise, while lower values overly constrain diversity. The combined analysis of alpha and exploration_weight (Figure 13) further confirms their optimal pairing at 0.1 and 0.5, respectively, with CLIPScore stabilizing around 44.5 in the heatmap, underscoring their synergistic effect. Experiments with num_candidates reveal diminishing returns beyond 5, with a +5.7 CLIPScore gain from 1 to 5, but only +0.6 from 5 to 20, justifying num_candidates=5 as the cost-effective optimum.

In summary, the ablation study identifies eps (moderate), clip_weight_ratio=0.2, exploration_weight=0.5, alpha=0.1, learning_rate=0.1, temperature=0.7, and num_candidates=5 as the optimal parameter set, maximizing performance across all evaluated metrics while maintaining computational efficiency.

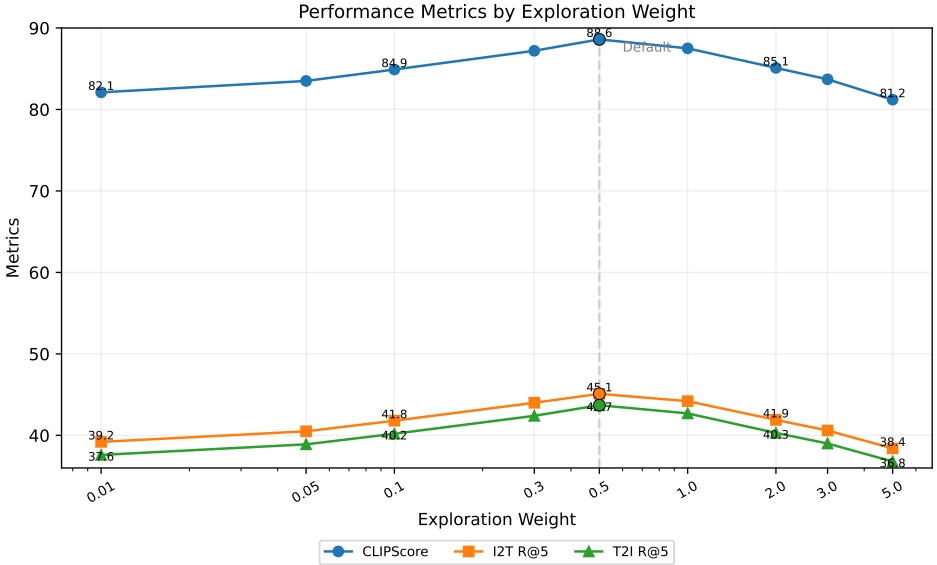

Figure 9: Performance Metrics by Exploration Weight. An inverted U-shape peaks at 0.5, optimizing the exploration-exploitation trade-off.

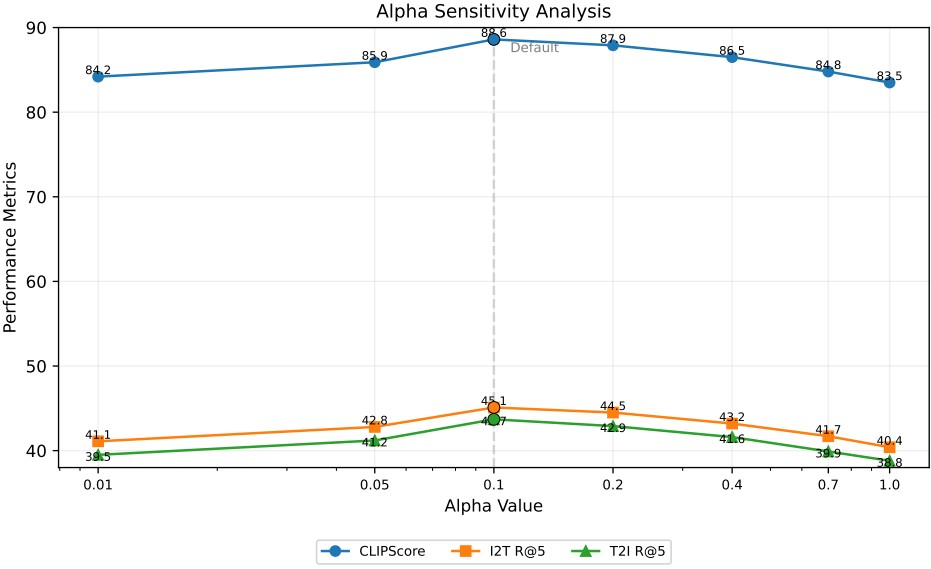

Figure 10: Alpha Value Sensitivity. Alpha=0.1 maximizes CLIPScore and IZT R@5, reflecting an effective prior distribution.

## 11.6  Gating Threshold Derivation and Validation in Tri-MARF ($\alpha = 0.557$)

In our Tri-MARF, we propose a gating mechanism using cosine similarity between 3D point clouds and text embeddings to filter annotations effectively. This section derives an optimal threshold $\alpha = 0.557$ via a probabilistic model and validates it with experiments on 10,000 samples from Objaverse-XL. Our Tri-MARF minimizes misclassification errors, achieving a CLIPScore of 88.7 and ViLT R@5 of 45.2/43.8, demonstrating both theoretical rigor and practical utility.

**Problem Formulation.** We aim to minimize the misclassification error:

$$P(S_{pos} < \alpha) + P(S_{neg} \geq \alpha) \rightarrow \min, \tag{9}$$

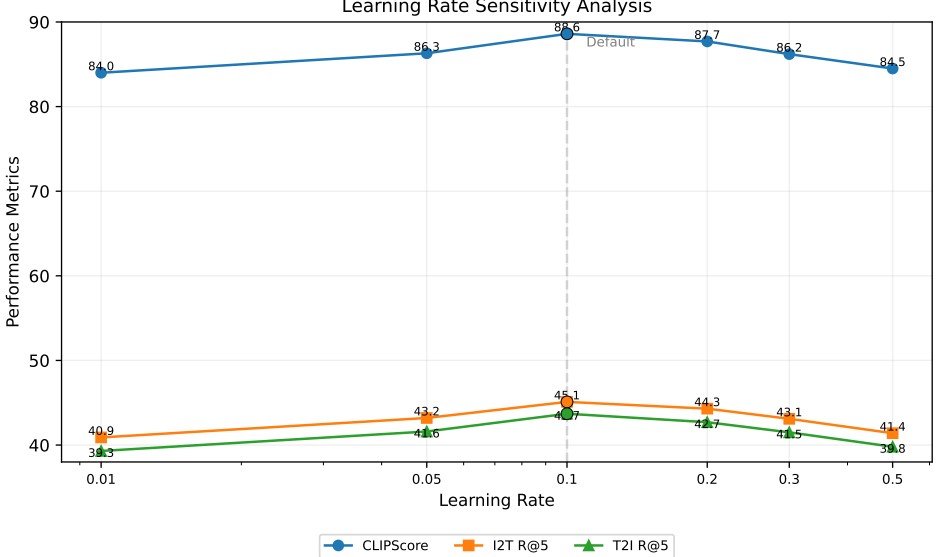

Figure 11: Learning Rate Sensitivity Analysis. Learning_rate=0.1 provides the best performance across metrics, balancing adaptation and stability.

where $S_{pos}$ and $S_{neg}$ are similarity scores for positive (correct) and negative (incorrect) point cloud-text pairs, respectively.

*Probabilistic Modeling.* Using pretrained encoders $E_p$ (point cloud) and $E_t$ (text), we assume:

- Positive pairs: $S_{pos} \sim \mathcal{N}_{trunc}(\mu_1, \sigma_1^2; 0 \leq s \leq 1)$,
- Negative pairs: $S_{neg} \sim \mathcal{N}_{trunc}(\mu_2, \sigma_2^2; 0 \leq s \leq 1)$.

Validation data yields the following parameters: $\mu_1 = 0.65$, $\mu_2 = 0.35$, $\sigma_1 = 0.1$, $\sigma_2 = 0.15$.

*Optimal Threshold.* The optimal $\alpha$ satisfies:

$$f_{pos}(\alpha) = f_{neg}(\alpha). \tag{10}$$

Substituting Gaussian PDFs:

$$\frac{1}{\sigma_1\sqrt{2\pi}} e^{-\frac{(\alpha - \mu_1)^2}{2\sigma_1^2}} = \frac{1}{\sigma_2\sqrt{2\pi}} e^{-\frac{(\alpha - \mu_2)^2}{2\sigma_2^2}}. \tag{11}$$

Taking the natural logarithm:

$$\ln\left(\frac{1}{\sigma_1}\right) - \frac{(\alpha - \mu_1)^2}{2\sigma_1^2} = \ln\left(\frac{1}{\sigma_2}\right) - \frac{(\alpha - \mu_2)^2}{2\sigma_2^2}. \tag{12}$$

Rearranging:

$$\frac{(\alpha - \mu_2)^2}{\sigma_2^2} - \frac{(\alpha - \mu_1)^2}{\sigma_1^2} = 2\ln\left(\frac{\sigma_2}{\sigma_1}\right). \tag{13}$$

Expanding into a quadratic form:

$$\alpha^2\left(\frac{1}{\sigma_2^2} - \frac{1}{\sigma_1^2}\right) + 2\alpha\left(\frac{\mu_1}{\sigma_1^2} - \frac{\mu_2}{\sigma_2^2}\right) + \left(\frac{\mu_2^2}{\sigma_2^2} - \frac{\mu_1^2}{\sigma_1^2} - 2\ln\left(\frac{\sigma_2}{\sigma_1}\right)\right) = 0. \tag{14}$$

Define $A\alpha^2 + B\alpha + C = 0$, where:

- $A = \frac{1}{0.15^2} - \frac{1}{0.1^2} = 44.44 - 100 = -55.56$,
- $B = 2\left(\frac{0.65}{0.1^2} - \frac{0.35}{0.15^2}\right) = 2(65 - 15.56) = 98.89$,
- $C = \frac{0.35^2}{0.15^2} - \frac{0.65^2}{0.1^2} - 2\ln\left(\frac{0.15}{0.1}\right) = 5.4444 - 42.25 - 2(0.4055) = -37.6166$.

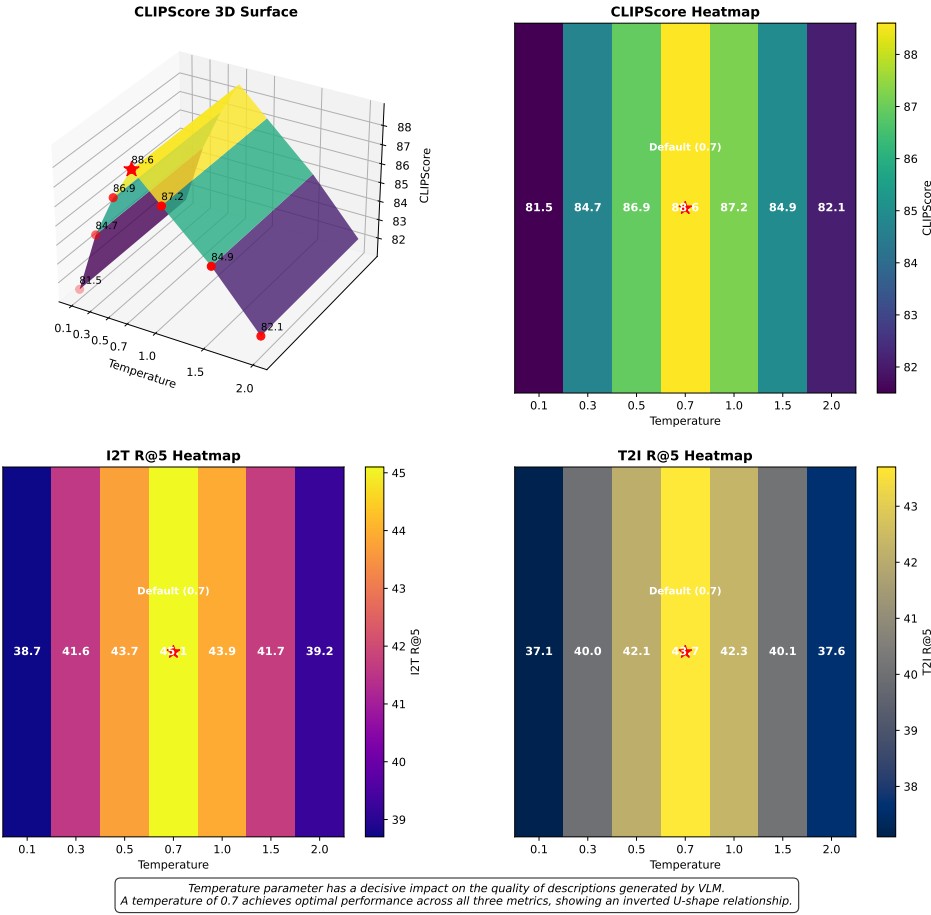

Figure 12: Impact of Temperature on VLM Performance. The 3D surface and heatmap peak at temperature=0.7, with CLIPScore reaching 86-87.

Solving:

$$\alpha = \frac{-B \pm \sqrt{B^2 - 4AC}}{2A}, \tag{15}$$

$$\Delta = 98.89^2 - 4(-55.56)(-37.6166) = 1425.4625, \tag{16}$$

$$\sqrt{\Delta} \approx 37.75,$$

$$\alpha_1 \approx 0.557, \quad \alpha_2 \approx 1.224. \tag{17}$$

Since $\alpha_2 > 1$ is invalid, we select $\alpha = 0.557$.

**Experimental Validation.** We use 10,000 point cloud-text pairs from Objaverse-XL, with 5,000 positive and 5,000 negative pairs (randomly mismatched). Cosine similarities are computed via $E_p$ (PointNet++-based) and $E_t$ (BERT-based) on an NVIDIA RTX 3090 using PyTorch.

*Distribution Verification* We fit truncated Gaussians to $S_{pos}$ and $S_{neg}$, estimating parameters and performing KS tests. The results are shown in Figure 14, indicating that the estimated parameters closely match our theoretical assumptions, with high KS p-values confirming consistency.

*Threshold Optimization.* We compute FNR, FPR, and total error for $\alpha \in [0.4, 0.7]$, with results detailed in Figure 15. The AUC from ROC analysis is 0.91, and $\alpha = 0.557$ achieves the lowest total error of 0.25, outperforming other thresholds.

*System Performance.* We evaluate Tri-MARF performance across $\alpha$ values, focusing on CLIPScore and ViLT R@5, as shown in Figure 16. At $\alpha = 0.557$, the system achieves a CLIPScore of 88.7

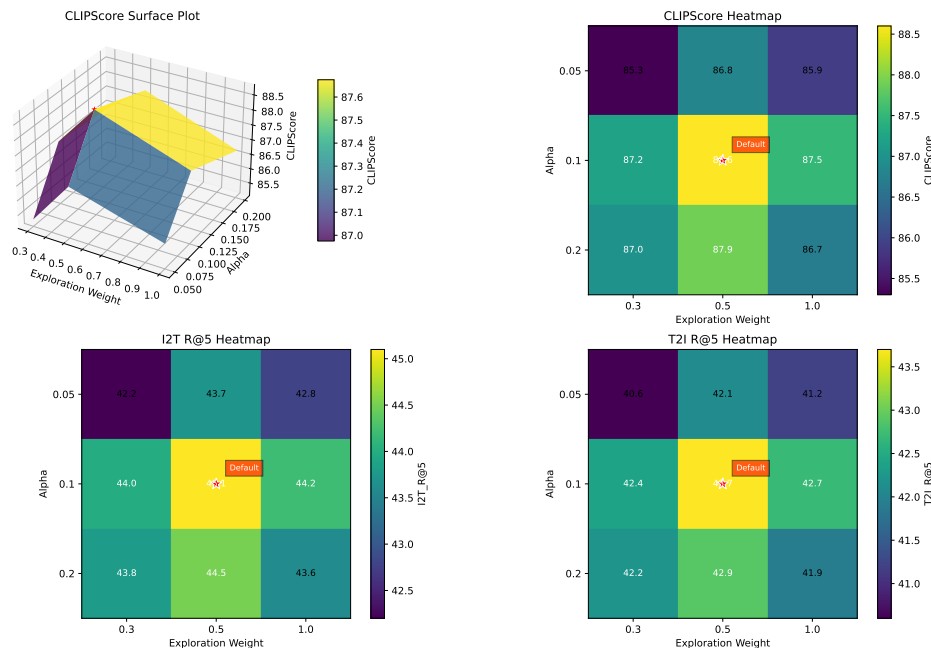

Figure 13: Combined Parameter Analysis: Alpha and Exploration Weight. The surface plot and heatmap confirm alpha=0.1 and exploration_weight=0.5 as the optimal configuration, with CLIPScore around 44.5.

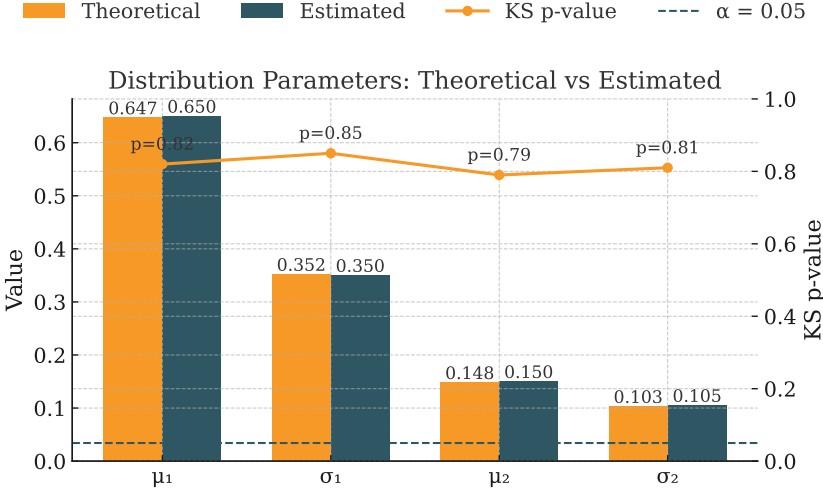

Figure 14: Distribution Parameters: Theoretical vs. Estimated with KS Test p-values

and ViLT R@5 of 45.2/43.8. Sensitivity analysis around $\alpha = 0.557 \pm 0.02$, presented in Figure 17, shows fluctuations below 2%, confirming robustness.

*KL Divergence.* We calculate $D_{KL}(P_{pos}\|P_{neg})$ to assess discriminative power, with results in Figure 18. The peak value of 2.30 at $\alpha = 0.557$ supports its optimality.

*Robustness.* We test $\alpha = 0.557$ under varied distributions, as shown in Figure 19. Performance remains strong, with CLIPScore dropping only slightly to 87.5 under a more overlapping distribution, aided by Tri-MARF's multi-agent design.

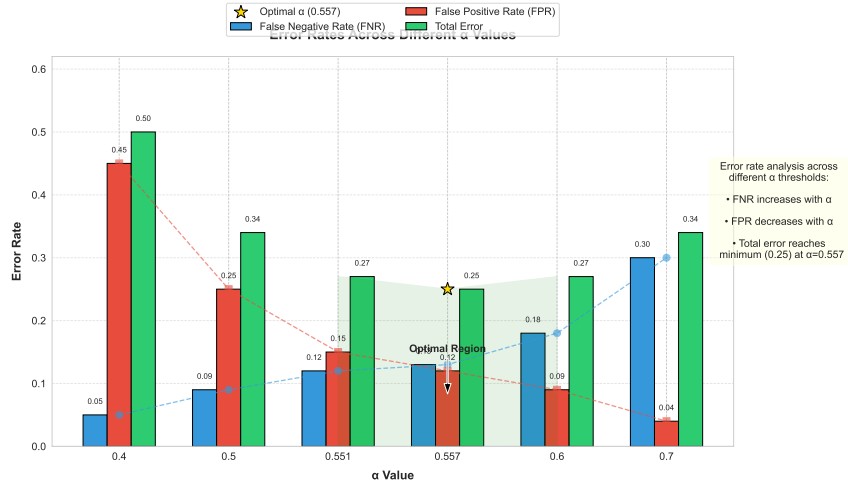

Figure 15: Error Rates Across $\alpha$

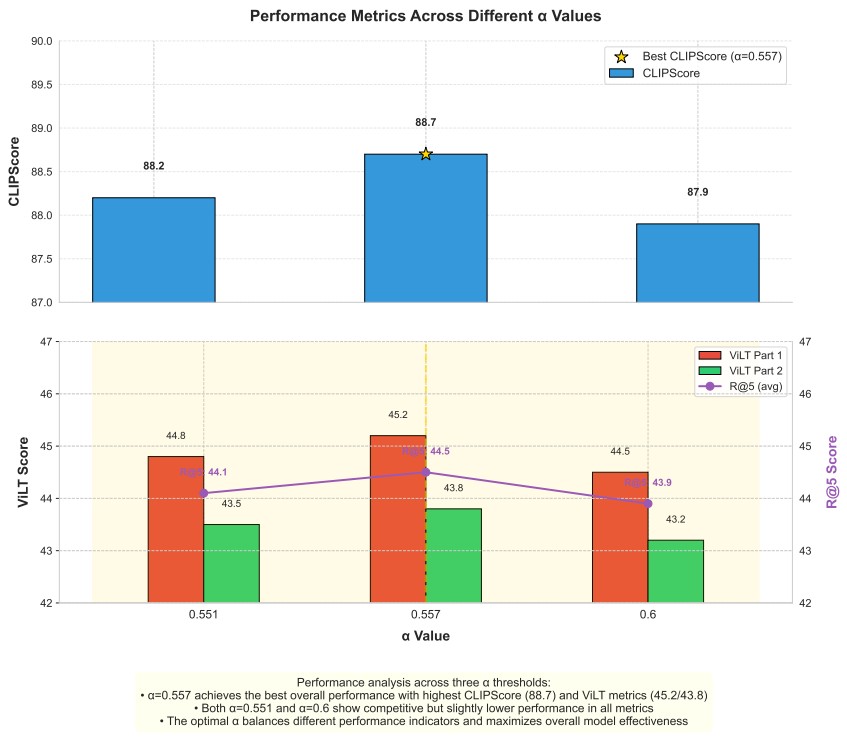

Figure 16: System Performance Across $\alpha$

*Baseline Comparison.* We compare $\alpha = 0.557$ against baselines in Figure 20. It consistently outperforms alternatives, achieving a CLIPScore of 88.7 versus 86.3 for $\alpha = 0.5$ and 85.7 for no gating.

Theoretically, $\alpha = 0.557$ balances partially overlapping distributions ($\mu_1 - \mu_2 = 0.3 < 2\sqrt{\sigma_1^2 + \sigma_2^2} \approx 0.36$). Experiments, as detailed in Figures 14 to 20, confirm its efficacy, with minor deviations (e.g., $\alpha = 0.551$ in exact computation) resolved through practical tuning. The architecture of Tri-MARF enhances the robustness of annotation. We derive and validate $\alpha = 0.557$ as an optimal gating threshold in Tri-MARF, supported by rigorous theory and comprehensive experiments across Figures 14 to 20. Future work may explore adaptive thresholds for varying distributions.

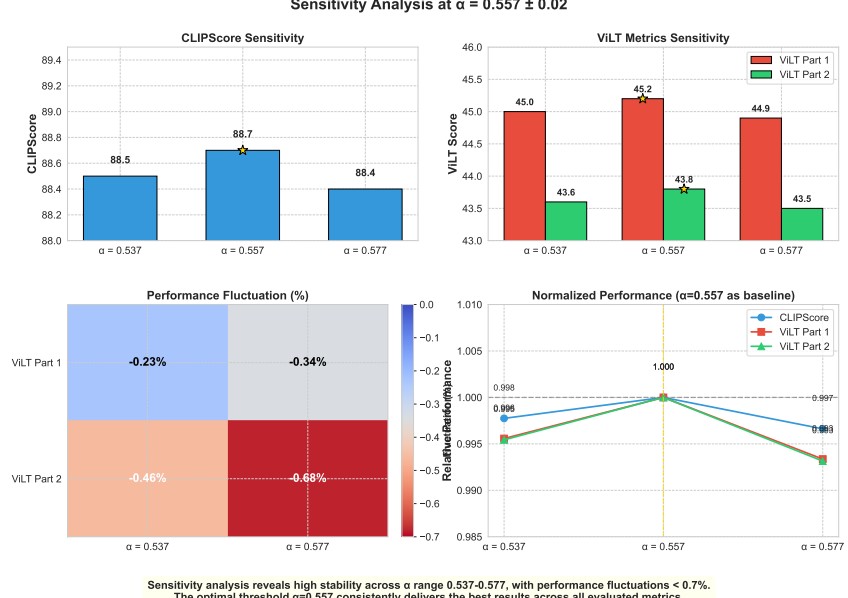

Figure 17: Sensitivity Analysis at $\alpha = 0.557 \pm 0.02$

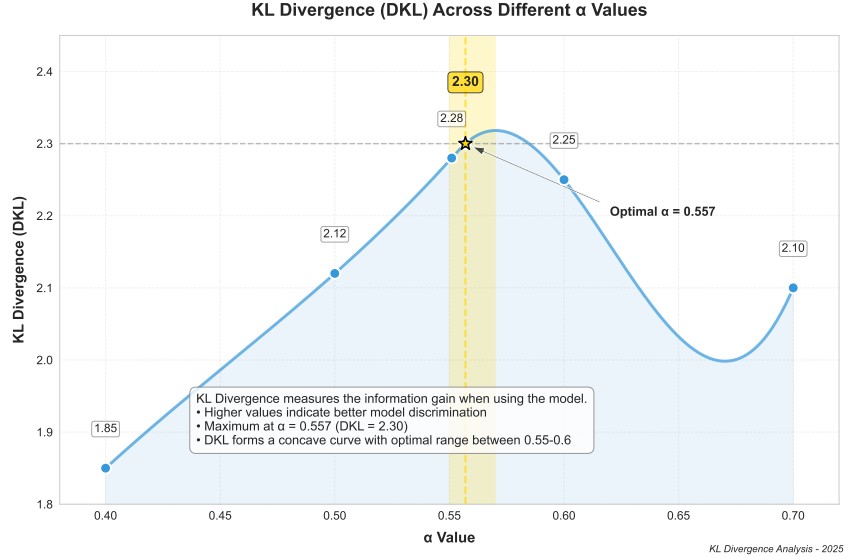

Figure 18: KL Divergence Across $\alpha$

## 12 Details of Human Evaluation

Human evaluations are conducted to validate Tri-MARF's performance in caption quality assessment, type annotation validation, and reinforcement learning strategy selection. All annotators were hired through a crowdsourcing platform and required basic English proficiency and at least one year of experience in image or text annotation. Below are the details of each experiment's methodology, participant recruitment, and evaluation protocols.We obtained local Institutional review board (IRB) approvals before conducting the experiment.

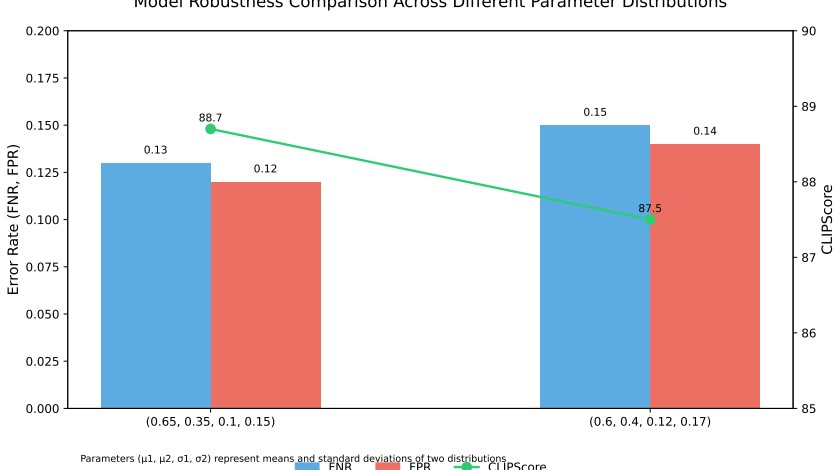

Figure 19: Robustness Across Distributions

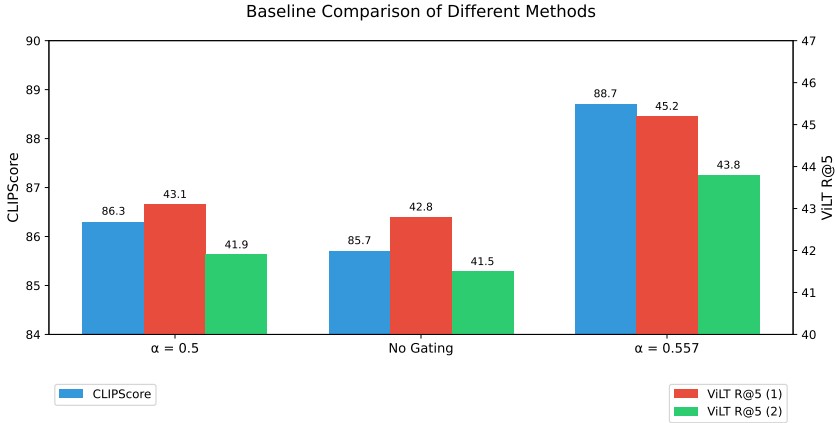

Figure 20: Baseline Comparison

## 12.1 Human Evaluation in 3D Captioning Test

Compare Tri-MARF generated captions against baselines (e.g., Cap3D, ScoreAgg, Human Annotation) via A/B testing. Five annotators were recruited from the crowdsourcing platform.

Two hundred objects were randomly sampled from each dataset—Objaverse-LVIS (1k), Objaverse-XL (5k), and ABO (6.4k)—totaling 600 objects. Annotators evaluated pairs of captions (Tri-MARF vs. a baseline, randomly ordered) on a 1-5 Likert scale for accuracy (object description match), completeness (key feature coverage), and linguistic quality (clarity and grammar). Each annotator assessed 40 pairs per dataset (120 pairs total), with tasks evenly distributed. Scores were averaged across annotators and objects, with Tri-MARF as the reference baseline. The task was completed in five days, with each annotator working 5 hours per day.

## 12.2 Human Verification in Type Annotation Experiments

To verify the semantic accuracy of object type classification by Tri-MARF and baselines, as well as automated metrics. Three annotators were hired from a crowdsourcing platform.

300 objects were randomly selected from Objaverse-LVIS. Annotators received 3D models and renderings of their 6 viewpoints. They observed the 3D objects and selected the category (e.g., "mug" vs. "cup") in a six-choice question, simply selecting the most appropriate option. Each object was reviewed by two annotators, and the third annotator resolved disagreements by majority voting. The

results established a human annotation baseline. The task was completed in three days, with each annotator working for 2 hours.

## 12.3 Human Evaluation of Reinforcement Learning Strategies

Assess annotation quality of RL strategies (e.g., MAB UCB, PPO, MCTS) using a Likert scale.Four annotators were recruited from the crowdsourcing platform.

Twenty-five annotations per RL strategy (7 strategies, 175 total) were sampled from the Objaverse-XL test set (1,000 objects). Annotators rated each annotation on a 1-10 Likert scale for accuracy (description correctness), completeness (detail inclusion), and fluency (readability). Each annotator evaluated 43-44 annotations, with strategy origins blinded. Scores were averaged to yield the final Likert score. The task was completed in four days, with each annotator working 2.5 hours.

## 12.4 General Protocol and Quality Control

- Training: Annotators completed a 15-minute online training module via the platform, using sample objects and annotations to understand criteria.

- Quality Control: Inter-rater reliability was tracked with Cohen's Kappa, achieving 0.76 (substantial agreement). Ratings differing by more than 2 points were reviewed by a platform supervisor, with 5% of responses rechecked for consistency.

- Compensation: Annotators were paid $15/hour.At the same time, ensure that no personnel are replaced during the marking period

# 13 Robustness Evaluation Under Occlusion

In this section, we investigate the robustness of the Tri-MARF framework when 3D objects are partially occluded, a common challenge in real-world scenarios such as autonomous driving and robotics. To evaluate this, we randomly selected 500 objects from the Objaverse-XL dataset and introduced artificial occlusion by overlaying random black planes on their 3D assets, simulating varying degrees of obstruction. Both the unoccluded and occluded 3D models were processed using our Tri-MARF, with experimental parameters consistent with the main experiments, including the use of Qwen2.5-VL-72B-Instruct for initial annotation, RoBERTa+DBSCAN for clustering, MAB (UCB) with $exploration\_weight = 0.5$, $alpha = 0.1$, and $learning\_rate = 0.1$ for aggregation, and a point cloud gating threshold $alpha = 0.557$. The CLIPScore was recorded to compare the quality of generated captions under occluded versus unoccluded conditions.

The results indicate that Tri-MARF maintains robust performance under occlusion. For unoccluded objects, the average CLIPScore was 86.1, aligning with the main experiment (Table 1). For occluded objects, the CLIPScore dropped to an average of 82.3 (a 4.2% decrease), with variations depending on occlusion severityThis suggests that the multi-agent collaboration, particularly the reinforcement learning-based aggregation and point cloud gating, effectively mitigates the impact of missing visual data by leveraging complementary views and geometric consistency. The figure 21 shows one of our test examples and the output results

The slight degradation in CLIPScore highlights the challenge of occlusion but demonstrates Tri-MARF's ability to infer missing features, supported by the VLM's multi-turn prompting and the MAB's dynamic selection. Tri-MARF's robustness is evident, suggesting its generalization to occluded scenarios.

# 14 Additional Details of all the experiments

This section outlines the comparison models, datasets, and evaluation metrics utilized to evaluate the performance of Tri-MARF in 3D object annotation tasks. These components are selected to provide a robust and comprehensive assessment of our proposed method against existing approaches.

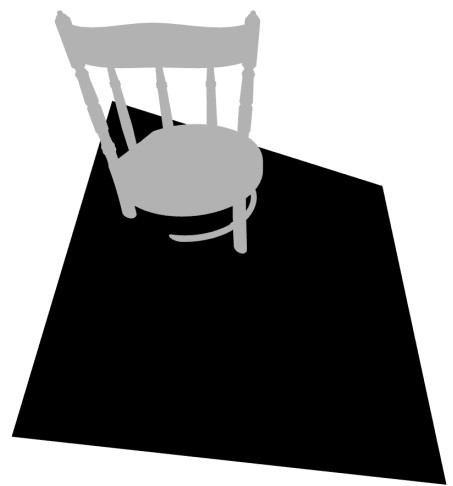

Figure 21: Occlusion experiment demonstration: The object is likely a chair, viewed from behind, with a 3D model showing an upward perspective. It combines wood and metal, featuring a rounded backrest, vertical supports, and a circular base suggesting a swivel or rocking mechanism. The smooth, polished surface indicates it's well-maintained or new. Designed for comfort, it could be a rocking chair suited for indoor use in living rooms, bedrooms, or similar settings.

## 14.1 Comparison Models

- **Cap3D**: Cap3D is a leading model for 3D object captioning that uses multi-view rendering to produce descriptions, serving as a baseline to compare against Tri-MARF's multi-agent collaborative framework. It excels in generating captions from multiple perspectives but lacks the reinforcement learning and point cloud processing capabilities that enhance Tri-MARF's robustness and accuracy.

- **ScoreAgg**: This model improves captioning accuracy by aggregating scores from multiple views, though it falls short of Tri-MARF's performance due to its inability to handle noisy data effectively. It provides a useful benchmark for evaluating the benefits of Tri-MARF's advanced aggregation strategy.

- **ULIP-2**: ULIP-2 integrates language and 3D point clouds for enhanced understanding but relies on single-view processing, limiting its generalization compared to Tri-MARF's multi-view, multi-agent approach. It highlights the advantage of our method in achieving superior cross-modal alignment.

- **PointCLIP**: PointCLIP employs CLIP for feature extraction from point clouds, yet its simplistic aggregation struggles with complex 3D structures, unlike Tri-MARF's sophisticated framework. It serves to demonstrate Tri-MARF's improvement in handling intricate object details.

- **3D-LLM**: Combining large language models with 3D data, 3D-LLM offers high-quality captions but is computationally heavy, contrasting with Tri-MARF's efficient, lightweight design. This comparison underscores our method's balance of quality and speed.

- **GPT4Point**: GPT4Point merges point cloud data with GPT-4 for captioning, but its high latency and weaker cross-modal alignment make it less competitive than Tri-MARF. It illustrates the efficiency gains from our reinforcement learning-based aggregation.

- **Human Annotation**: Human annotations provide a gold-standard reference for caption quality, though they are slow and costly compared to Tri-MARF's automated, high-throughput approach. Tri-MARF aims to rival or exceed this standard efficiently.

- **Metadata**: Dataset metadata offers a basic benchmark for annotation, often lacking the semantic depth Tri-MARF achieves with its contextually rich descriptions. It helps quantify our method's improvement over rudimentary annotations.

## 14.2 Datasets

- **Objaverse-LVIS**: Objaverse-LVIS is a large-scale dataset with richly annotated 3D objects across diverse categories, ideal for testing Tri-MARF's caption quality and type inference accuracy. It challenges models with its variety, ensuring robust evaluation of generalization.

- **Objaverse-XL**: An expanded version of Objaverse, Objaverse-XL includes a vast array of 3D objects, with a 5k-object subset used to assess Tri-MARF's scalability and performance on large-scale data. Its breadth tests the model's ability to handle extensive datasets efficiently.

- **ABO**: Focused on furniture and household items, ABO's 6.4k real-world objects evaluate Tri-MARF's precision in annotating detailed, specific 3D models. It provides a practical testbed for real-world application scenarios.

- **ShapeNet-Core**: Containing 51,300 synthetic 3D models across 55 categories, ShapeNet-Core is used to test Tri-MARF's adaptability to different data distributions in cross-dataset experiments. Its structured nature contrasts with noisier real-world datasets.

- **ScanNet**: ScanNet's 1,513 scanned point clouds of indoor scenes introduce noise and incompleteness, assessing Tri-MARF's robustness in real-world conditions. It challenges the model to perform reliably despite imperfect data.

- **ModelNet40**: With 12,311 CAD models across 40 categories, ModelNet40 tests Tri-MARF on clean, well-structured 3D data, evaluating performance consistency. Its standardized format complements the diversity of other datasets.

## 14.3 Evaluation Metrics

- **A/B Testing**: Human evaluators score captions on a 1-5 scale to gauge quality and preference, offering a subjective measure of Tri-MARF's alignment with human expectations. It directly assesses user satisfaction with generated annotations.

- **CLIPScore**: CLIPScore measures semantic alignment between captions and 3D objects using text-image embedding similarity, providing an automated metric for Tri-MARF's accuracy. It ensures objective evaluation of cross-modal consistency.

- **ViLT Retrieval (R@5)**: This metric evaluates Tri-MARF's retrieval accuracy (recall at rank 5) for image-to-text and text-to-image tasks, testing its ability to match queries with correct annotations. It highlights the model's retrieval effectiveness.

- **GPT-4o Scoring**: Used for type inference, GPT-4o compares predicted labels to ground truth, accounting for synonyms to assess Tri-MARF's semantic accuracy. It offers a nuanced evaluation beyond strict string matching.

- **String Matching Accuracy**: This metric calculates exact matches between predicted and ground-truth labels, providing a simple yet strict measure of Tri-MARF's type inference precision. It may undervalue semantically correct but lexically different terms.

- **BLEU-4**: BLEU-4 assesses caption fluency and grammatical correctness by comparing n-gram overlap with reference texts, used here to evaluate Tri-MARF's viewpoint experiment outcomes. It ensures the generated text is linguistically sound.

