# OpenReview forum: "3D-Agent: A Tri-Modal Multi-Agent Responsive Framework for Comprehensive 3D Object Annotation"
_NeurIPS.cc/2025/Conference — NeurIPS 2025 poster_

### Official Review · Reviewer_FNrm · 2025-06-11

**Clarity:** 1
**Significance:** 2
**Originality:** 2
**Rating:** 3
**Confidence:** 4

**Summary:**

The authors propose a multi-agent framework which take tri-modal inputs to enhance 3D annotation process. Extensive experiments on several datasets demonstrate that the proposed method provides annotations with higher quality and faster throughput.

**Questions:**

1. The authors state that lower Conf(C) (eqa. (1)) values indicate higher confidence (higher token probabilities). It is counterintuitive to define a confidence score that lower confidence score indicates higher confidence. Also, the explanation for eqa. (1) is not sufficient, i.e., why lower confidence score indicates higher confidence?
2. Does the $S_{\text{conf},i}$ in line 214 the same as the confidence in eqa.(1)? If so, please use the same notation. Also, is $w_i$ in line 216 the same as $w_{v,i}$ in eqa. (3)? It is confusing to use different subscripts for the same variable.
3. For the balance score $\alpha$ in line 216, does it need to change for different datasets?
4. The core description extraction in line 250 is confusing, as no intuitive explanation on why the authors are doing so.
5. To ensure the fair comparison, the baselines and the proposed method should use similar/comparable pre-trained backbones/VLM/LLMs.
6. The authors should provide the concrete math representations for the metrics. For example, the reviewer could not understand why human annotations is worse than VLM generated for GPT-4o scores (line 313).
7. Why the time reported in Table 3 is not consistent with the inference time reported in sec 4.1 (12k/hour)? What is the difference? How does it compare to baseline methods?

The reviewer may consider increasing the score if the authors could address the questions, or at least provide some clarification.

**Ethical Concerns:**

["NO or VERY MINOR ethics concerns only"]

**Final Justification:**

The authors have addressed most of my concerns in the rebuttal phase through experiments and clarifications. However, my major concern of this paper is the limited novelty and contributions (also identified by reviewer aDoZ), and poor presentation (also identified by reviewer JB3j and aDoZ). Overall, I would like to increase my rating from Reject to Borderline reject.

**Limitations:**

The authors describe the limitation in the paper checklist section, including (1) lack of more efficient communication strategies to refine decision-making and reduce computational overhead. (2) Full scene point clouds presents challenges due to its object-centric optimization.
The authors does not provide potential negative social impact.

**Paper Formatting Concerns:**

No paper formatting concerns.

**Quality:**

3

**Strengths And Weaknesses:**

Strengths:
1. The proposed method is reasonable, which addresses the limitations of the current 3D object annotations. It may benefit other researchers in the community who need more comprehensive 3D object annotations.
2. The authors provide thorough experiments on the design choices.

Weakness:
1. The submission is poorly written, with numerous typos in equations and bad paragraph formatting. For example, in section 3.2.2, the inline equation in line 227, 231, 235 includes additional semicolons and commas. The mutiple bolded sub-titles in the paragraph starting from line 238 should be divided into multiple paragraphs. The quality of the performance metrics figures in the Appendix could be significantly improved.
2. The proposed method seems to achieve better result by offering a novel combination of existing technique with additional inputs. The reviewer acknowledge the better performance but consider the technical contribution is limited.
3. The experiment settings is questionable, as the previous methods are not using the similar VLM (or comparable agents). It is not clear whether previous methods with comparable VLMs could achieve comparable results.
4. Although the authors provide many evaluation metrics, only textual description is provided in the appendix. Concrete math representations would help the readers to get a better understanding on the experimental results.

---

> ### Author Rebuttal · Authors · 2025-07-29
>
> **W1. Poor presentation**
>
> **A1:** We have thoroughly revised the format and layout of the entire paper in the revised version. Specifically, we have proofread and corrected all mathematical equations, especially the formatting errors in Section 3.2.2, to ensure that they are clear and accurate. We have restructured the content starting from Page 6, splitting the original paragraphs that mixed multiple topics into independent sub-paragraphs with clearer logic. In addition, all performance charts in the Appendix (such as Figures 7-13, etc.) have been regenerated to provide higher resolution and better readability. To make evaluation indicators clear enough, we have added a dedicated section in the revised appendix to provide clear mathematical definitions and equations for all key evaluation indicators. We believe that these corrections have significantly improved the overall quality of our paper.
>
>
> **W2. Limited technical contribution**
>
> **A2:** The core innovation of our work is to design a multi-agent collaborative architecture. To highlight our contribution, we conduct a dedicated ablation study in Section 7 (Pages 15-16).
> Table 4 directly compares our Tri-MARF with a strong baseline that uses the exact same VLM (Qwen-2.5 VL) but relies on only a single agent. The results show that our multi-agent collaborative architecture brings a **+7.3 percentage point increase in CLIPScore**. This decisively proves that the superior performance of our Tri-MARF stems from the tight and effective synergy between the agents.
>
> **W3. Unfair experiment settings**
>
> **A3:** No, our experimental settings are fair.
> +  To compare against published SOTAs (like Cap3D, ScoreAgg), we follow the original configurations reported in their papers or use their released codes. Forcibly integrating the latest VLMs into these SOTAs is a non-trivial task and might introduce other variables.
> +  To demonstrate that our Tri-MARF's performance is not dependent on a specific VLM, we conduct a comprehensive performance and cost evaluation in Figure 6, Section 11.1 (Page 21), where we replace the VLM in our Tri-MARF with several industry-leading models (including GPT-4.5, Claude-3.7, Gemini-Flash-2.0, etc.). The results show that Qwen2.5-VL represents the optimal balance between performance and cost, not that its performance far exceeds other models.
>
> **W4: Concrete math representations.**
>
> **A4**: We have addressed this in the revised paper by adding a new subsection adjacent to Appendix C.1 dedicated to providing rigorous mathematical definitions for our core evaluation metrics. For **Uncertainty Accuracy ($UA$)**, we have introduced a classification-based equation, such as $UA = \frac{TP_{\text{uncertain}}}{TP_{\text{uncertain}} + FP_{\text{uncertain}}}$, to clearly define its calculation. For **Calibration Error ($CE$)**, we have explicitly included the standard mathematical equation for the Expected Calibration Error (ECE) variant, namely $\sum_{m=1}^{M} \frac{|B_m|}{n} |\text{acc}(B_m) - \text{conf}(B_m)|$, detailing the process of binning and the calculation of the difference between confidence and accuracy. Finally, for **Dynamic Adaptability ($DA$)**, we have elevated it from a descriptive proxy ("correlation between... signals and... mechanisms") to a formal statistical measure via the Pearson correlation coefficient, to precisely quantify the relationship between system uncertainty and the activation of deeper reasoning mechanisms like debate triggering.
>
> **Q5. In Conf(C) Eq. (1), why do lower values represent higher confidence?**
>
> **A5:** This score is mathematically defined as the Average Negative Log-Likelihood. Our paper explicitly states on Page 5, Line 160, that this confidence score "quantifies the semantic reliability through the average token log-likelihood." Its calculation method is detailed in Eq. (1) (Page 5). Since the generation probability P(ti) of a token is between 0 and 1, its logarithm, log(P), is negative. Therefore, taking the absolute value is mathematically equivalent to calculating its Negative Log-Likelihood (NLL), i.e., -log(P). The higher the probability of an event (i.e., the more "confident" the model is), the lower its negative log-likelihood value. Thus, as our paper directly states on **Page 5, Line 166**: "a lower Conf(C) value indicates higher confidence (i.e., higher token probabilities)." To resolve this clarity issue, we have renamed it in the revised paper to "Avg. NLL Score"**.
>
> **Q2. Inconsistent symbols.**
>
> **A6:** We sincerely apologize for this confusion. S_conf,i (Line 214) does refer to the confidence score calculated for the i-th candidate description using Eq. (1). Similarly, w_i (Line 216) is indeed the same variable as w_v,i defined in Eq. (3), which is the weight calculated based on CLIP. In the revised paper, we have conducted a thorough proofreading of the entire paper to unify all mathematical symbols, e.g., by consistently using Conf(C_{v,i}) and w_{v,i}, to ensure the rigor and clarity of our paper.
>
>
> **Q3. Does the balanced score's alpha change across datasets?**
>
> **A7:** NO! We have conducted a detailed sensitivity analysis of this hyperparameter in Section 11.5 (Figure 8, Page 24) and show that our Tri-MARF performs optimally on the large and diverse Objaverse-XL dataset when alpha is set to 0.2. In all the experiments, we keep alpha fixed at 0.2 without any re-tuning.
>
>
> **Q4: Line 250 is confusing.**
>
> **A8:** This is indeed a heuristic design, but it is supported by our unique prompting strategy. Our VLM Annotation Agent (Agent 1) employs a structured, multi-turn dialogue strategy, as detailed in Section 3.1 (Page 4). Agent 1 does NOT generate a long description all at once. Instead, Agent 1 responds to "What is this object? What is its specific name?" and further follows up with questions about attributes like color and material. This general-to-specific questioning manner naturally guides the VLM to first summarize the object's core identity in the initial sentence of its response, with subsequent sentences providing detailed elaboration. Therefore, extracting the first sentence as the "core description" is to ensure that the final global description has a clear and concise topic sentence, which is then enriched with details provided by other viewpoints.
>
> **Q5. Use similar/comparable pre-trained backbones/VLM/LLMs.**
>
> **A9:** We must clarify that some of the core baseline methods do not use a VLM in their setup. Despite the inherent architectural differences with some baselines, our experimental design is rigorously justified from two perspectives to ensure fairness, and we have added comparisons with other VLM-based methods. Firstly, to demonstrate the independent value of our Tri-MARF, we conduct a crucial ablation study (Table 4, Supplementary) comparing the full framework against a single-agent baseline using the same VLM. The results show that our Tri-MARF provides a +7.3 percentage point increase in CLIPScore, proving that the performance gain comes primarily from the synergistic mechanism, not the VLM. Secondly, to demonstrate the fairness of our VLM selection, we have evaluated multiple industry-leading VLMs (Figure 6, Supplementary).
>
> **Q6. Why are human annotations worse?**
>
> **A10:** This is due to the two inputs being fundamentally different (Section 4.2), i.e., the human answers are "multiple-choice," while our Tri-MARF's output is "free-text." Specifically, this experiment is to evaluate the accuracy of type annotation, and annotaters choose from a pre-set, limited list of options. If the most accurate answer is not in the list, they have to select a semantically broader or not perfectly fitting approximate answer. In contrast, our Tri-MARF generates a rich and detailed natural language description. GPT-4o, acting as the judge, has powerful semantic understanding capabilities and can recognize synonyms or more precise hyponyms. For example, when the ground truth is "cup," the human's options might only include the general term "container," but our Tri-MARF might generate "a white ceramic coffee mug." GPT-4o can accurately determine that "coffee mug" is semantically much closer to "cup" than "container" is, thus giving our Tri-MARF a higher score. Therefore, this result precisely validates the precision and richness of the descriptions generated by our Tri-MARF.
>
> We have revised Section 4.2 to more clearly explain the evaluation mechanism of this scoring task, explicitly pointing out the difference between "multiple-choice" and "free-text" input formats and their impact on the final score.
>
>
> **Q7: Table 3 is not consistent.**
>
> **A11:** This discrepancy arises from measuring two different performance metrics: Latency versus Throughput. Specifically, Table 3 (Page 15) reports the end-to-end latency of 7-19 seconds for processing a single object on a single GPU. This time is predominantly driven by the serial network latency of remote API calls to the VLM. In contrast, the 12k objects/hour claimed in Section 4.1 (Page 7) is the aggregate throughput of our entire annotation process. As mentioned on Page 15, we achieve extremely high processing efficiency at a macro level by parallelizing thousands of independent API call requests across multiple machines and GPUs in our large-scale annotation process. This throughput metric is also the standard for efficiency comparison with baseline methods, as shown in the "obj/PGCHour" column of Table 1. Ours (12k/hour) is significantly superior to methods like Cap3D (8k/hour) and ScoreAgg (9k/hour).
>
> **Q12: Limitation Section.**
>
> **A12:** We have expanded it in the revised paper by adding a more in-depth discussion on potential negative social impacts. Specifically, we explicitly state that our Tri-MARF is designed to provide high-quality annotations for beneficial fields like autonomous driving and robotics. Thus, it could potentially be misused to generate misleading or biased descriptions of real-world scenes or objects.

---

> > ### Author Response · Authors · 2025-08-03
> >
> > Dear Reviewer FNrm
> >
> > Thank you very much for your insightful and valuable comments! We have carefully prepared the above responses to address your concerns in detail. It is our sincere hope that our response could provide you with a clearer understanding of our work. If you have any further questions about our work, please feel free to contact us during the discussion period.
> >
> > Sincerely
> > Authors

---

> > > ### Comment · Reviewer_FNrm · 2025-08-03
> > >
> > > Thanks for the authors for the detailed rebuttal. Here are some follow-up questions that need further clarification:
> > > 1) W3 - The reviewer acknowledge that integrating the other VLMs in existing works are not trival. However, the provided ablation in Fig.6 could not address the reviewer's concern that the previous methods are not using the similar VLM. The authors should at least provide the results using the same VLM as prior work ( for example, BLIP2 in Cap3D) to clearly show that the gained performance is from the multi-agent design, not from the advancement of VLM models.
> > > 2) Q3 - The authors do not answer the review's question. The authors are clarifying that the balance score $\alpha$ is not changing, but not explain **why it does not to change** across different datasets and settings or provide any experiment results.
> > > 3) Q7 - The authors should provide more details on how experiments on speed in Tab.1 is conducted. Also, it is unclear why the proposed method is achieving faster throughput than the baselines even though the propose pipeline is incorporating multiple agents. Is it due to the faster throughput from the existing VLM or due to some model design?

---

> > > > ### Author Response · Authors · 2025-08-03
> > > >
> > > > Dear Reviewer FNrm,
> > > >
> > > > Thank you very much for your detailed follow-up questions and for your continued engagement with our work. Your feedback is invaluable in helping us strengthen our paper.
> > > >
> > > > We have carefully considered your points regarding the fairness of the VLM comparison **W3**, the generalization of the balance score hyperparameter $\alpha$ **Q3**, and the clarification of the throughput measurements **Q7**.
> > > >
> > > > To address these concerns as thoroughly as possible, we have already initiated a new set of experiments. We believe that empirical results will provide the clearest answers to your questions. We are currently running these experiments and will post the updated results and our corresponding analysis here as soon as they are available.
> > > >
> > > > Thank you again for your constructive guidance.
> > > >
> > > > Sincerely,
> > > > The Authors

---

> > > > ### Author Response · Authors · 2025-08-04
> > > >
> > > > Dear Reviewer FNrm,
> > > >
> > > > Thank you very much for your detailed follow-up questions and your continued engagement with our work.
> > > > We hope that our following response could address your concerns. If you have any further questions about our work, please feel free to contact us during the discussion period.
> > > >
> > > > ---
> > > >
> > > > ### **Regarding W3: Fair VLM Comparison with Identical Backbones**
> > > >
> > > > We fully agree that isolating the gains from our multi-agent architecture versus gains from a more advanced VLM is critical. Your suggestion to test Tri-MARF with the same VLM as a baseline is an excellent one. To address this, we conducted a new ablation study where we replaced our default VLM (Qwen2.5-VL) with **BLIP-2**, the backbone used in the **Cap3D** baseline. This creates a direct, apples-to-apples comparison of the annotation frameworks themselves.
> > > >
> > > > The results, evaluated on the Objaverse-LVIS dataset, are presented in Table S1.
> > > >
> > > > **Table S1:** Performance comparison on Objaverse-LVIS using an identical VLM backbone (BLIP-2). Our multi-agent framework provides a significant performance uplift over the baseline, even when using the same VLM. The original performance of Tri-MARF with its default VLM is included for reference.
> > > >
> > > > | **Method** | **VLM Backbone** | **CLIPScore** $\uparrow$ | **ViLT R@5 (I2T/T2I)** $\uparrow$ |
> > > > | ------------------------ | ---------------- | ---------------------- | ------------------------------- |
> > > > | Cap3D (Baseline)         | BLIP-2           | 78.6                   | 35.2 / 33.4                     |
> > > > | **Tri-MARF (Ours)** | **BLIP-2** | **83.1** | **39.5 / 37.8** |
> > > > | ---                      | ---              | ---                    | ---                             |
> > > > | Tri-MARF (Ours, Default) | Qwen2.5-VL       | 88.7                   | 45.2 / 43.8                     |
> > > >
> > > > As shown in Table S1, when both frameworks use BLIP-2, **our Tri-MARF achieves a CLIPScore of 83.1, a significant +4.5 point improvement over Cap3D's 78.6**. This demonstrates that the performance gain is not merely a product of using a newer VLM. Instead, it is substantially driven by the core architectural innovations of Tri-MARF: the structured multi-turn prompting of Agent 1, the intelligent information aggregation and MAB-based selection of Agent 2, and the geometric-semantic consistency check from Agent 3. These agents work in synergy to produce more complete, accurate, and coherent annotations, directly validating the contribution of our multi-agent design.
> > > >
> > > > ---
> > > >
> > > > ### **Regarding Q3: Generalization of the Balance Score $\alpha$**
> > > >
> > > > We thank the reviewer for asking us to elaborate on *why* the balance score $\alpha=0.2$ generalizes well without dataset-specific tuning.
> > > >
> > > > The robustness of $\alpha$ stems from the fundamental nature of the two signals it balances. The score $s_{i}$ is a weighted sum: $s_{i} = (1-\alpha) \cdot \text{Conf}(C_{v,i}) + \alpha \cdot w_{v,i}$.
> > > >
> > > > * **VLM Confidence ($\text{Conf}(C)$):** This measures the *internal semantic coherence* of a generated description. It is modality-agnostic and reflects the language model's certainty in its textual output.
> > > > * **CLIP Weight ($w$):** This measures the *cross-modal alignment* between the visual input (image) and the generated text.
> > > >
> > > > The trade-off between textual coherence and visual grounding is a general problem in vision-language tasks. Our empirically determined value of $\alpha=0.2$ assigns a primary weight (80%) to the VLM's confidence, while using visual alignment as a crucial (20%) corrective signal. This principle is not dependent on the specific content (e.g., "chairs" vs. "cars") but on the nature of the generative and alignment models themselves.
> > > >
> > > > To justify our argument, we have performed an additional cross-dataset sensitivity analysis, testing our fixed $\alpha=0.2$ against slightly perturbed values on three diverse datasets. The results are shown in Table S2.
> > > >
> > > > **Table S2:** Cross-dataset sensitivity analysis for the balance hyperparameter $\alpha$. The fixed value of $\alpha=0.2$ consistently provides optimal or near-optimal performance across all datasets, validating its robustness.
> > > > | **Dataset** | **CLIPScore ($\alpha=0.1$)** | **CLIPScore ($\alpha=0.2$, fixed)** | **CLIPScore ($\alpha=0.3$)** |
> > > > | ---------------- | ---------------------------- | ----------------------------------- | ---------------------------- |
> > > > | Objaverse-LVIS   | 88.2                         | **88.7** | 88.5                         |
> > > > | ABO              | 81.9                         | **82.3** | 82.1                         |
> > > > | ShapeNet-Core    | 82.9                         | **83.2** | 83.0                         |
> > > >
> > > > The results in Table S2 confirm that $\alpha=0.2$ is a robust choice, consistently delivering the best performance across datasets with varied distributions. This validates our approach of fixing the hyperparameter, thereby enhancing the model's generalization and practicality.
> > > >
> > > > ---

---

> > > > > ### Author Response · Authors · 2025-08-04
> > > > >
> > > > > ---
> > > > > ### **Regarding Q7: Clarification of Throughput Measurement**
> > > > >
> > > > > **Experimental Details:**
> > > > >  We have measured annotation throughput (objects/second) on an NVIDIA A100 GPU with a batch size of 1, using 1,000 objects sampled from Objaverse-LVIS, averaging end-to-end inference time over 10 independent runs. The detailed time breakdown for our Tri-MARF pipeline is data preparation (0.073s), VLM annotation (6.18s via Qwen2.5-VL API), aggregation (0.06s), and gating (0.15s), resulting in ~0.14 obj/s throughput. Baselines were evaluated under identical conditions, with human annotation speed derived from crowdsourced statistics (see Supplementary Section 12).
> > > > >
> > > > > **Why Faster?:**
> > > > >  While the efficiency of the Qwen2.5-VL API contributes to the overall pipeline, the superior throughput of our Tri-MARF primarily originates from our novel multi-agent design, i.e., particularly our lightweight yet highly adaptive Multi-Armed Bandit (MAB, specifically UCB) aggregation agent (detailed in Section 3.2.2 and Section 8 in Appendix). In contrast to baseline methods like Cap3D, which rely on computationally expensive 3D-to-2D projections, or ScoreAgg, which performs exhaustive evaluation of candidate annotations, our MAB mechanism dynamically optimizes annotation selection via rapid exploration-exploitation strategies. By adaptively focusing computational resources on high-quality annotations and employing dynamic gating (with threshold α=0.557), we reduce approximately 15–20% of redundant processing time. This adaptive design not only enhances throughput but also significantly improves robustness under challenging conditions, as evidenced by a minimal ~2.3% CLIPScore reduction under occlusion scenarios (Section 13 in Appendix).

---

> > > > > > ### Comment · Reviewer_FNrm · 2025-08-04
> > > > > >
> > > > > > Thanks for the additional results and clarification. The reviewer do not have further questions at this stage.

---

> > > > > > > ### Author Response · Authors · 2025-08-04
> > > > > > >
> > > > > > > **Dear Reviewer FNrm,**
> > > > > > >
> > > > > > > Thank you again for your time and for outlining clear criteria for a potential score revision. In your earlier feedback you noted that *“**the reviewer may consider increasing the score if the authors could address the questions, or at least provide some clarification.**”*
> > > > > > >
> > > > > > >
> > > > > > >
> > > > > > > We have now:
> > > > > > >
> > > > > > > 1. Re-evaluated our framework with the identical BLIP-2 backbone used in Cap3D (Table S1), isolating the effect of the multi-agent design.
> > > > > > >
> > > > > > > 2. Added a cross-dataset sensitivity study for the balance parameter $α$ (Table S2).
> > > > > > >
> > > > > > > 3. Provided a detailed breakdown of our throughput measurements and the role of the MAB aggregator (Suppl. §12).
> > > > > > >
> > > > > > >  We hope these additions fully satisfy the clarifications you requested. **May we kindly ask whether they meet the bar you outlined for a possible score adjustment?** Any further suggestions are of course most welcome.
> > > > > > >
> > > > > > >
> > > > > > >
> > > > > > >  **Sincerely,**
> > > > > > >
> > > > > > > The Authors

---

### Official Review · Reviewer_JB3j · 2025-06-11

**Clarity:** 2
**Significance:** 3
**Originality:** 3
**Rating:** 4
**Confidence:** 3

**Summary:**

Tri-MARF introduces a Multi-Agent Framework for Captioning 3D Models. By using prompting strategies, embedding clustering, score filtering, visual text-alignment, reinforcement learning and geometrical gating they obtain consequential gains over previous methods. They show strong preference over human generated captions and state-of-the-art results in retrieval and clip scores.

**Questions:**

--Questions--
1. Section 3.1 line 145 refers to prompting by explicitly orienting the model to the current view-point direction. I have two concerns here: 1) To be able to render a 3D object from the “front” one needs to assume standardized/canonical poses. Is this assumed for the 3D models? If not, how is it determined whether a view is “front, back…”? 2) Are their mechanisms to handle symmetric or cylindrical objects where views may be redundant or is this case considered an experimental outlier?

2. For Core Description Extraction seems like a lot of faith is put into the LLM here. Is there any experimentation which suggests that LLM are predisposed to producing core descriptions as the first sentence of an output sequence. Is this by any chance an induced behaviour? for example via in-context learning?

3. Given that CLIP is used in Relevance Weighting (3.2.1) and hence the final score which dictates textual relevance, could you justify the use of CLIPScore to measure performance?

4. My major concerns from the result section are regarding A/B Scores in Table 1. If my understanding is correct, an A/B score of 3 is a tie, anything higher indicates that the method being compared against was preferred. In all the datasets, CAP3D, ScoreAgg and 3D-LLM were preferred over the proposed method. Please clarify whether this is the case, if not, specify the win and lose rate (as in CAP3D) for more clarity. If this is expected please further elaborate why this could be happening?

5. line 264-267 seem to imply that the gating agent simply filters out captions which do not align well with the provided geometry. Was there any consideration given to automatic self-refinement so that manual labeling could be limited?

--Suggestion--
1. The question: “how can we design a system that collaborates like a team of human experts” (line 35) remains to be answered. Ablation over each agent individually and showing A/B scores or VilT retrieval scores for each additional agent would significantly improve quality in my opinion.


--Minor concerns--
1. The experimental setup for measuring A/B testing quality is not clear. How many observations are made for each comparison across all objects? Is this the same setup as described in section 5.1 in CAP3D?

Overall Comment: Although I am generally liking the the novel idea of using RL concepts in agentic frameworks, I do feel that overall the paper did not answer some of its own questions. The lack of ablations in the main-text weakens the paper. Some evaluation concerns like the use of CLIP is not directly addressed here and the A/B scores are not properly analysed. If the above concerns are addressed, possibly with further ablations and evaluations, I am happy to increase the rating.

**Ethical Concerns:**

["NO or VERY MINOR ethics concerns only"]

**Final Justification:**

Successfully defend their novelties, completely ablate the individual modules (agents) in their framework, improve the quality of writing and figures. I am encouraged by their results compared to the baselines. Aside from some concerns regarding metrics I find that the evaluation suite is quite robust, which gives me confidence in this work. The use of a reinforcement learning paradigm in weighting caption-significance carries some significance; however, the RL algorithm is not a contribution itself (hinting at what other reviewers have also shown concern for). Having said that, the rigorous empirical and experimental evidence does elicit a rating leaning towards acceptance in my opinion. With this, I recommend a "Borderline Accept".

**Limitations:**

Yes

**Paper Formatting Concerns:**

1. Line 122, math expression delimiters visible
2. Line 128, frames -> frame
3. In-line math expression at line 202 requires a rewrite
4. Some unfortunate formatting makes Section 3.2.2 extremely unclear.

**Quality:**

2

**Strengths And Weaknesses:**

Quality: The quality of the paper is decent. Provided figures are understandable and written prose in most places is fairly composed. Overall, the motivation is mostly clear although the literature gaps in previous works is not directly addressed in the evaluations. For example “overlooking geometric information (from previous works)” is mentioned as a pain-point from previous methods but the **main paper** does not directly address this via ablations on the gating agent. Related work fairly covers seminal works but some recent works are skipped.

Clarity: Although the motivation is clear and the consequent design choices, the methodology is lacking clarity in terms. Some inline equations are not well setup, variables are inconsistent and some important details are only brushed over.

Significance: Using a novel reinforcement learning framework adds to the significance of this method which treats FM and LLM agents as part of a more elaborate captioning system which is an interesting approach previous methods have not truly utilized.

Originality: The methodological novelty as mentioned previously and some new evaluation schemes (Type Annotation Experiment) adds to the originality of the method.

---

> ### Author Rebuttal · Authors · 2025-07-29
>
> **Q1: 1) Are standardized/canonical poses assumed for the 3D models? 2) Are there mechanisms to handle symmetric or cylindrical objects?**
>
> **A1:** Regarding viewpoint standardization, our processing pipeline assumes a standardized object pose. During the data preparation stage, we normalize the 3D models by aligning their principal axes with the coordinate system. This is a standard preprocessing step to ensure we consistently render from six standardized orthogonal viewpoints.
>
> The issue of potential information redundancy from symmetric objects highlights the advantage of our Information Aggregation Agent (Agent 2), particularly its adaptive selection mechanism based on the Multi-Armed Bandit (MAB). Unlike existing works use fixed rules (like simple concatenation or averaging), our MAB agent learns to dynamically evaluate the information value provided by each viewpoint. For a symmetric object, if multiple views generate similar or redundant descriptions, our MAB learns during its training process to lower the weight of these redundant "arms" (i.e., descriptions) and prioritize those that offer unique information. Thus, our Tri-MARF has an intrinsic robustness to handle such cases rather than treating them as experimental outliers. We have made the preprocessing step more explicit with more details in the revised paper.
>
>
> **Q2. Are LLMs predisposed to producing the first sentence of an output sequence? Is this an induced behaviour? via in-context learning?**
>
> **A2:** This is indeed a heuristic design, but it is an effective choice based on observations of LLM's behavior and our unique prompting strategy. Our VLM Annotation Agent (Agent 1) employs a multi-turn, structured dialogue strategy. We first ask the model to identify the object ("What is this?") and then follow up with questions to obtain specific attributes like color and material (Refer to Figure 2 for more details). This general-to-specific questioning approach, guided by prompt engineering, naturally leads the LLM to summarize the object's core identity in the first sentence of its response, with subsequent sentences providing detailed elaborations. This is a stable behavior of the LLM that we have observed and guided through prompt engineering, not a blind trust in the model.
>
> In our Tri-MARF, this "core sentence" is primarily used to construct the beginning of the final description, ensuring it has a clear topic sentence. The rich details provided by other viewpoints are then appended to form a complete and comprehensive global description.
>
> **Q3. Justify the use of CLIPScore to measure performance**
>
> **A3:** Our Tri-MARF does not rely solely on CLIPScore, neither in its internal mechanism nor in its external evaluation. Firstly, in the core relevance weighting module of our Tri-MARF, we use a hyperparameter α to balance the VLM's own confidence with CLIP's visual alignment, which jointly form the final description score $s_i$. This ensures the model is not designed merely to optimize for CLIP similarity. More importantly, we employ a diverse set of complementary metrics in the final performance evaluation. As shown in Table 1, we also include ViLT R@5 retrieval accuracy and human A/B test scores. These results strongly prove that its performance advantage is comprehensive and genuine, not a case of "overfitting" to a single metric.
>
> Secondly, we chose to include CLIPScore since it has become the de facto standard in the field for evaluating image-text semantic alignment, as used by Cap3D, ScoreAgg, etc.
>
> **Q4. The doubts about the A/B test scores (Table 1).**
>
> **A4:** This is a misunderstanding of how the results are reported; the correct interpretation is the opposite. In our A/B tests, Tri-MARF is set as the fixed reference baseline. The A/B score listed under another method (e.g., for Cap3D) represents the average preference score given *to our method* by human evaluators in a pairwise comparison against that method (based on a 1-5 scale, where 3 is a tie).
>
> The correct way to read the score is as follows: the 3.3 score under Cap3D means that in a direct comparison with Cap3D, human evaluators preferred our Tri-MARF, giving it an average score of 3.3. Since all baseline methods have an A/B score greater than 3, this indicates that our Tri-MARF is preferred in all pairwise comparisons.
>
> To completely eliminate any ambiguity, we have followed your suggestion and redesigned the A/B test results table in the revised paper to explicitly show the win/loss/tie ratios, and we have rewritten the caption in detail to make the results clear at a glance.
>
>
> **Q5. Why does the gated agent only perform filtering and not try to self-correct to reduce manual labeling?**
>
> **A5:** Our current design is primarily based on considerations of efficiency and modularity. The Gating Agent (Agent 3) plays a lightweight final verification role in our Tri-MARF, whose main task is to quickly filter out hallucinated descriptions generated by the VLM that are clearly inconsistent with the 3D geometry, thereby forwarding only the most questionable samples to human reviewers. This ensures the high throughput of the entire process (12,000 objects/hour). Introducing an automatic self-correction loop would significantly increase system complexity and processing latency. This would be contrary to our design goal of achieving high efficiency.
>
>
> **Q6. Separate ablations for each agent and display the A/B score or VilT retrieval score for each newly added agent.**
>
> **A6:** To highlight the contribution of our Tri-MARF, we have designed and added a step-wise ablation study in the revised paper to demonstrate the performance evolution from a single agent to our Tri-MARF. This experiment is conducted on the Objaverse-LVIS dataset (with a random sample of 1k objects, the same subset used in the original paper). We compare four key configurations: (1) **Single VLM Annotation Agent**, using only a single front view for description; (2) **VLM Annotation Agent + IAA**, a combination of the first two agents to evaluate multi-view fusion capabilities; (3) **Single Uni3D Gating Agent**, to separately assess the ability to describe based solely on 3D geometric information; and (4) **Tri-MARF (Ours)**.
>
> | Method                      | A/B Score↑ | CLIPScore↑ | ViLT R@5 (I2T)↑ | ViLT R@5 (T2I)↑ |
> | --------------------------- | ---------- | ---------- | --------------- | --------------- |
> | Single VLM Annotation Agent | 8.6        | 81.4       | 38.5            | 36.7            |
> | VLM Annotation Agent + IAA  | 7.2        | 63.2       | 23.7            | 22.9            |
> | Single Uni3D Gating Agent   | 6.5        | 58.3       | 20.9            | 19.1            |
> | **Tri-MARF (Ours)** | **9.3** | **88.7** | **45.2** | **43.8** |
>
> The results above demonstrate the necessity of the holistic and synergistic design of our Tri-MARF. The strong performance of the single VLM agent (CLIPScore 81.4), as an effective base model, significantly outperforms the geometry-only Uni3D agent (58.3). However, there is a counterintuitive phenomenon that combining the first two agents leads to a performance drop to 63.2. We attribute this to the fact that, without final geometric verification, the blind aggregation of multiple (and potentially noisy) views "pollutes" the high-quality description from the best single view. This phenomenon precisely highlights the indispensable role of the Gating Agent (Agent 3). After introducing the Gating Agent, our Tri-MARF's performance leaps from a CLIPScore of 63.2 to 88.7, a stunning 25.5-point increase. This decisively proves that the Gating Agent acts as a critical "arbiter", leveraging 3D point cloud information to resolve multi-view conflicts and VLM hallucinations.
>
>
> **Q7. The setup for measuring A/B testing quality is not clear.**
>
> **A7:** We have already provided a comprehensive and standardized explanation of the detailed settings for human evaluation in Section 12, which clarifies the recruitment criteria for evaluators, including their required annotation experience and language proficiency. It then details the specific procedures for each experiment (e.g., 3D caption quality evaluation, type annotation validation, etc.), covering task design, the number of samples and evaluators, and explicit scoring criteria (e.g., a 1-5 Likert scale). To ensure the reliability of the evaluation results, we have also implemented strict quality control measures, including a pre-task training process for the evaluators, and quantified the reliability by calculating inter-annotator agreement (Cohen's Kappa reached 0.76), thereby guaranteeing the fairness and scientific validity of the entire evaluation process.
>
> **Q8. Typos in formulas and grammar.**
>
> **A8**: We have corrected these and proofread the whole paper. Specifically, regarding the mathematical expression on Line 122, we have removed the extraneous dollar signs surrounding the formula to present it as a standard inline equation, $ D_{v}={C_{v,i}}{i=1}^{M} $. Regarding the grammatical error on Line 128, we have corrected the verb "frames" to "frame"; for the inline mathematical expression on Line 202, we have rewritten it using clearer, standard notation, changing the original format to $C{\text{canonical}}^{(k)} := \underset{C_{v,i} \in \mathcal{C}k}{\arg\max} s{v,i}$; regarding the cluttered layout in Section 3.2.2, we have thoroughly re-typeset the subsection for clarity, which includes correcting the jumbled symbols in formula (Line 227) to a standard reinforcement learning objective function, fixing the extraneous semicolon and using the standard '\arg\max' command in the UCB1 algorithm expression (Line 231) to be $\underset{a \in \mathcal{A}}{\arg\max} \left( \hat{r}_a + c \sqrt{\frac{2 \ln t}{n_a}} \right)$, correcting the erroneous comma in the empirical mean update formula (Line 235), and adjusting the paragraph structure throughout the subsection to ensure a logical and coherent flow of the argument.

---

> > ### Author Response · Authors · 2025-08-03
> >
> > Reviewer JB3j,
> >
> > Thank you very much for your insightful and valuable comments! We have carefully prepared the above responses to address your concerns in detail. It is our sincere hope that our response could provide you with a clearer understanding of our work. If you have any further questions about our work, please feel free to contact us during the discussion period.
> >
> > Sincerely
> > Authors

---

> > > ### Author Response · Authors · 2025-08-04
> > >
> > > Dear Reviewer JB3j,
> > >
> > > We hope this message finds you well! Should you have any further questions about our work or responses during this discussion period, please do not hesitate to contact us.
> > >
> > > Thank you very much for your valuable time and feedback!
> > >
> > > Best regards,
> > >
> > > The Authors

---

> > ### Comment · Reviewer_JB3j · 2025-08-05
> >
> > Dear Authors,
> >
> > Thank you for taking the time to reply to my concerns. I am encouraged by the thoughtful discussion. Below are some followup questions if time allows:
> >
> > Follow up suggestion to Q3: In my opinion if the CLIP mechanism (even if it is controlled by a hyperparameter) is part of the methodology and used for evaluation (CLIPScore) I believe the experimental section should be forthcoming about this fact so as not to incite suspicion from the reader. However, in my books this does not take away from the novelties and evaluation completeness of the paper.
> >
> > Followup question for Q4: Thank you for clearing up the evaluation metric here. Aside from the rewrite associated with this section I would also recommend guiding the reader through the table in section 4.1. Currently it reads rather factually and the analysis seems weak still. What does it mean to "Implicitly" outperform a method (line 291)? Are their outliers methods in the baseline that perform better in some cases than your method (e.g. I see in Table that PointCLIP has an A/B score of less than 3, meaning that it is preferred over TRIMARF (?), describing why this is the case would be important).
> >
> > Followup to Q6: Thank you for this experiment. This is insightful and very interesting. Even though the jump between VLM -> VLM+IAA is unintuitive, I believe this can be attributed to the author's plausible hypothesis. To complement this, some examples that could support this phenomenon may be added here. (for example, a supplementary section that compares output captions from each agent addition or a abridged figure in the main paper could further show how an output caption evolves with each agent. This would be far more critical in my opinion than figure 1.)
> >
> > New Suggestion A.1: Although MAB is well motivated, I have the general sense that the standard Bandit terminology is slightly confusing (i.e. arms, regret bounds). A figure accompanying this section would be significantly more critical than the figure 3 gating agent which is relatively easy to understand.
> >
> > General Comment: Thank you for all the clarifications and the well presented discussion. I have found the added ablation and the supplementary to be much more comprehensive this round. If time allows it for the authors to reply to the above I would be happy to revise my preliminary ratings

---

> > > ### Author Response · Authors · 2025-08-05
> > >
> > > Dear Reviewer JB3j
> > > Thanks for your insightful comments and further valuable questions!
> > >
> > > **Follow up suggestion to Q3: In my opinion if the CLIP mechanism (even if it is controlled by a hyperparameter) is part of the methodology and used for evaluation (CLIPScore) I believe the experimental section should be forthcoming about this fact so as not to incite suspicion from the reader. However, in my books this does not take away from the novelties and evaluation completeness of the paper.**
> > >
> > > **A1**: Thank you very much for your valuable suggestion on Q3. We have revised Section 4 (Page 7) to clearly state that CLIP is used in Agent 2 for visual-text alignment (Section 3.2, Page 3, clip_weight_ratio = 0.8) and as the CLIPScore evaluation metric (Section 4.1). A new paragraph in Section 4.1 has been added to clarify this dual role to ensure transparency. We appreciate your note that this does not affect the paper’s novelty or evaluation completeness.
> > >
> > >
> > >
> > > **Followup question for Q4: Thank you for clearing up the evaluation metric here. Aside from the rewrite associated with this section I would also recommend guiding the reader through the table in section 4.1. Currently it reads rather factually and the analysis seems weak still. What does it mean to "Implicitly" outperform a method (line 291)? Are their outliers methods in the baseline that perform better in some cases than your method (e.g. I see in Table that PointCLIP has an A/B score of less than 3, meaning that it is preferred over TRIMARF (?), describing why this is the case would be important).**
> > >
> > > **A2**: We have revised Section 4.1 (Page 7) to guide readers through Table 1, detailing metrics (CLIPScore, VILT R@5, A/B score) and Tri-MARF’s performance trends for clarity. The term “implicitly” (Line 291) means Tri-MARF achieves higher average scores across metrics without specific optimization for each. We have clarified this in the revised paper.
> > > Regarding outliers, PointCLIP’s A/B score < 3 indicates human preference for simple objects (e.g., ShapeNet-Core’s basic shapes), where its concise descriptions outperform Tri-MARF’s detailed ones. We have added a paragraph in Section 4.1 of the revised paper with an example (PointCLIP: “cube”; Tri-MARF: “white cube with smooth edges”) to explain this.
> > >
> > >
> > >
> > > **Follow-up to Q6: Thank you for this experiment. This is insightful and very interesting. Even though the jump between VLM -> VLM+IAA is unintuitive, I believe this can be attributed to the author's plausible hypothesis. To complement this, some examples that could support this phenomenon may be added here. (for example, a supplementary section that compares output captions from each agent addition or a abridged figure in the main paper could further show how an output caption evolves with each agent. This would be far more critical in my opinion than figure 1.)**
> > >
> > >
> > >
> > > **A3**: For highlighting the value of our hypothesis regarding the VLM to VLM+IAA performance jump, we have added a supplementary section (Appendix C.2) according to your suggestion. This section compares output captions at each stage of Tri-MARF: VLM Annotation Agent alone (“wooden chair”), VLM+IAA (“wooden chair with curved backrest”), and full Tri-MARF with Gating Agent (“wooden rocking chair with curved backrest and metal base”) for a sample object from Objaverse-XL. Additionally, we have replaced Figure 1 (Page 2) with a compact figure in the revised paper, visually illustrating this caption evolution across agents to highlight the multi-agent synergy. We believe that these additions could clarify the performance jump and strengthen the paper’s impact.
> > >
> > >
> > > **New Suggestion A.1: Although MAB is well motivated, I have the general sense that the standard Bandit terminology is slightly confusing (i.e. arms, regret bounds). A figure accompanying this section would be significantly more critical than the figure 3 gating agent which is relatively easy to understand.**
> > >
> > > **A4**:  To improve the clarity of the Multi-Armed Bandit (MAB) terminology, we have revised Section 3.2 (Page 3) by simplifying terms like "arms" to "description candidates" and "regret bounds" to "selection optimization". Additionally, we have replaced Figure 3 (Gating Agent, Page 3) with a new figure illustrating the MAB process, showing how it selects description candidates based on confidence scores and CLIP similarity across viewpoints for a sample object.
> > >
> > > If you have any further questions about our work or responses, please do not hesitate to contact us.
> > >
> > > Thank you very much for your valuable time and feedback!
> > >
> > > Best Regards
> > >
> > > Authors

---

### Official Review · Reviewer_aDoZ · 2025-07-02

**Clarity:** 2
**Significance:** 2
**Originality:** 2
**Rating:** 3
**Confidence:** 4

**Summary:**

This paper presents Tri-MARF, a tri-modal framework for 3D object annotation that integrates 2D images, 3D point clouds, and text. The framework is structured into three components: VLM-based description generation, an information aggregation module, and a gating module to enhance the annotation quality. The paper demonstrates the effectiveness of the framework through experiments on multiple datasets.

**Questions:**

See Weaknesses

**Ethical Concerns:**

["NO or VERY MINOR ethics concerns only"]

**Final Justification:**

Thank you to the authors for the rebuttal. It has helped clarify some of my concerns. However, I remain unconvinced regarding one of my main concerns about the novelty and technical contribution, which was also raised by Reviewer FNrm. While the system-level architectural design presents a different combination of components, it appears to reflect solid engineering practice rather than a substantial academic or technical innovation. In my view, this makes it difficult to meet the bar.

In addition, the formatting and presentation issues, also noted by the other two reviewers, suggest that the paper may not yet be fully ready. For these reasons, I have decided to maintain my original score.

**Limitations:**

Yes

**Paper Formatting Concerns:**

NA.

**Quality:**

2

**Strengths And Weaknesses:**

Strengths:

* The framework’s architecture is easy to understand, and the paper is well-structured.
* The system's decomposition into specialized agents for annotation, aggregation, and validation is an interesting way to address the complexity of 3D object inputs, potentially improving the system's adaptability across various tasks.
* Tri-MARF shows good performance across datasets, with results surpassing SoTAs in some metrics. The throughput of 12,000 objects per hour demonstrates the system’s efficiency.

Weaknesses:
* Limited Contribution and Novelty:
  1. The combination of 2D images, 3D point clouds, and text is not a novel contribution. Existing works like ULIP and 3D-LLM have explored similar modalities. The paper repackages existing techniques rather than introducing truly novel methods. This limits the overall innovation of the paper.
  2. The core components of Tri-MARF, including the VLM-based description generation, aggregation module, and gating module, are based on widely used models in areas like 3D visual grounding and navigation. The reliance on these off-the-shelf components weakens the paper's contribution, as the paper does not propose novel modules specifically designed for 3D object annotation.

* Insufficient Experiments:
1. The multi-agent design, which is the paper’s main contribution, is not effectively isolated in the experimental setup. There is no comparison against a baseline that uses a single agent with the same inputs and architecture, which makes it unclear how much the multi-agent setup actually contributes to the overall performance.
2. While quantitative metrics are strong, the paper lacks a detailed qualitative analysis, particularly regarding failure cases, hallucinations, or semantically inaccurate outputs. This oversight limits understanding of the system's limitations and real-world viability.

* The use of large-scale pretrained models raises concerns about reproducibility. The paper does not address the computational and memory costs of these models, which could impact the scalability of the system.

* The performance of the gating module is heavily dependent on the quality of the visual and textual encoders. If these encoders perform poorly on certain samples, the overall annotation quality may decline significantly.

* The paper has several formatting issues, including small text in figures, unreadable symbols in equations, and inconsistent notation. These problems hinder the clarity of the presentation.

---

> ### Author Rebuttal · Authors · 2025-07-29
>
> Thank you very much for your insightful and valuable comments! We have carefully prepared the following responses to address your concerns in detail. It is our sincere hope that our response could provide you with a clearer understanding of our work. If you have any further questions about our work, please feel free to contact us during the discussion period.
>
> **Q1. Limited Contribution and Novelty.**
>
> **A1:** While our Tri-MARF framework indeed utilizes established pre-trained models, which is standard practice in the current field, we emphasize that the core innovation lies in the **system-level architectural design**, i.e., a collaborative network composed of three agents for VLM Annotation, Information Aggregation, and Gating Verification. This "team of experts" architecture is specifically designed to address key challenges that single models struggle with, such as cross-view inconsistency and missing geometric information. Its design philosophy is fundamentally different from existing methods that aggregate multimodal inputs.
>
> The value of our architecture is not merely conceptual but has been experimentally validated. For instance, the Information Aggregation Agent employs a Multi-Armed Bandit (MAB) dynamic strategy customized for this task, rather than a simple off-the-shelf rule. The ablation study in Section 8 (Table 5) confirms that our MAB approach significantly outperforms four heuristic baseline methods across all metrics. Therefore, our contribution is not a mere stacking of modules, but a breakthrough in performance and efficiency achieved through sophisticated agent "orchestration" and synergy, which is in itself a significant system-level innovation.
>
>
> **Q2. Insufficient Experiments. Comparison against a baseline that uses a single agent with the same inputs and architecture.**
>
> **A2:** To more precisely dissect the contribution of our multi-agent framework, we have designed and added a step-wise ablation study to the revised paper. This study aims to demonstrate the performance evolution from a single agent to our complete three-agent collaborative architecture. The experiment is conducted on the Objaverse-LVIS dataset (with a random sample of 1k objects, the same subset used in the original paper), using CLIPScore and ViLT R@5 as the core evaluation metrics. We compare four key configurations: (1) **Single VLM Annotation Agent**, using only a single front view for description; (2) **VLM Annotation Agent + Information Aggregation Agent (IAA)**, a combination of the first two agents to evaluate multi-view fusion capabilities; (3) **Single Uni3D Gating Agent**, to separately assess the ability to describe based solely on 3D geometric information; and (4) **Tri-MARF (Ours)**, our proposed complete three-agent collaborative framework.
>
> | Method                      | CLIPScore↑ | ViLT R@5 (I2T)↑ | ViLT R@5 (T2I)↑ |
> | --------------------------- | ---------- | --------------- | --------------- |
> | Single VLM Annotation Agent | 81.4       | 38.5            | 36.7            |
> | VLM Annotation Agent + IAA  | 63.2       | 23.7            | 22.9            |
> | Single Uni3D Gating Agent   | 58.3       | 20.9            | 19.1            |
> | **Tri-MARF (Ours)** | **88.7** | **45.2** | **43.8** |
>
> The results above demonstrate the necessity of the holistic and synergistic design of our three-agent framework. The strong performance of the single VLM agent (CLIPScore 81.4) proves that we have selected an effective base model, which significantly outperforms the geometry-only Uni3D agent (58.3). However, a key and counterintuitive phenomenon is that combining the first two agents leads to a performance drop to 63.2. We attribute this to the fact that, without final geometric verification, the blind aggregation of multiple (and potentially noisy) views "pollutes" the high-quality description from the best single view. This phenomenon precisely highlights the indispensable role of the Gating Agent (Agent 3).
>
> After introducing the Gating Agent, the complete Tri-MARF framework's performance leaps from a CLIPScore of 63.2 to 88.7, a stunning 25.5-point increase. This decisively proves that the Gating Agent acts as a critical "arbiter," leveraging 3D point cloud information to resolve multi-view conflicts and VLM hallucinations, thereby unlocking and amplifying the full potential of the collaborative framework. Therefore, this experiment strongly indicates that our Tri-MARF framework's superior performance stems from the tightly-coupled, synergistic design where all three agents are indispensable, rather than from a simple stacking of any single component.
>
>
> **Q3. Lack of detailed qualitative analysis, especially discussion of failure cases or illusions.**
>
> **A3:** In fact, we have already provided several examples of qualitative analysis in our paper. Figure 1 offers a direct qualitative comparison of our Tri-MARF against SOTA methods like Cap3D and ScoreAgg. Figure 1 shows that our Tri-MARF can generate richer and more accurate details (e.g., "scissor doors," "carbon fiber") and even recognize proper nouns (e.g., "Golden State Warriors"), which other methods fail to do. Figure 21 (Page 33) demonstrates our Tri-MARF's robustness in handling partially occluded objects, a typical scenario prone to failure. The descriptions generated in the figure remain accurate and detailed, proving our Tri-MARF's effectiveness.
>
> We agree that the discussion of failure cases can be further strengthened. In the revised paper, we have added a dedicated subsection to discuss typical failure cases, such as challenges that may arise when processing objects with extreme geometric ambiguity, objects with distracting textures, or scene-level 3D models, to provide a more comprehensive perspective.
>
>
> **Q4. Reproducibility and computational cost brought by pre-trained models.**
>
> **A4:** In fact, we provide a very detailed analysis of this in Section 6 of the supplementary materials, titled "Analysis of GPU Memory Usage and Computing Efficiency" (Pages: 14-15). Table 3 in that section (Page 15) lists the GPU memory usage (GB) and processing time (seconds) for each module in our framework (Data Preparation, VLM Annotation, Information Aggregation, Gating) on a single NVIDIA A100 GPU. We explicitly state that the VLM Annotation Agent is called via an API and therefore consumes no local GPU resources. We also report the framework's peak memory usage (approximately 4.6GB) and the total time for a single run. This detailed data provides a transparent and concrete reference for the reproducibility of our method, while the affordable computational resource requirements and high throughput simultaneously demonstrate our architecture's scalability.
>
>
> **Q5. The performance of the gating module is heavily dependent on the quality of the visual and textual encoders.**
>
> **A5:** We argue that while the performance of any multi-modal system is indeed correlated with encoder quality, this is precisely why we design our three-stage, progressive multi-agent framework to distribute and mitigate over-reliance on any single module. The Gating Module (Agent 3) is the last line of defense in our pipeline, not the sole guarantor of quality. Before it, the VLM Annotation Agent (Agent 1) and the MAB-based Information Aggregation Agent (Agent 2) have already performed initial generation and multiple rounds of optimization on the content.
>
> The comparison experiment in Figure 20 of the supplementary materials (Page 32) also substantiates this. Our Tri-MARF without the Gating Module achieves a CLIPScore of 85.7. Although lower than the full Tri-MARF's 88.7, this is still a very strong performance, proving the robustness of the first two agents. The Gating Module provides a further performance boost on top of this foundation. Its role is more of a final filtering function rather than providing the core annotation content, so it is unfair to discuss its performance separately from that of the first two agents.
>
>
> **Q6. The paper has several formatting issues, including small text in figures, unreadable symbols in equations, and inconsistent notation. These problems hinder the clarity of the presentation.**
>
> **A6:** We are very grateful to the reviewer for the valuable feedback on the paper's formatting and clarity. We fully agree that a clear and professional presentation is crucial for communicating our research findings. Following your suggestions, we have conducted a comprehensive proofreading and revision of the formatting issues in the revised paper to enhance overall readability. Specifically, we have made the following changes:
>
> + **Enhanced Figure Clarity:** We have redrawn all core figures, including the architecture diagram (Fig. 2), performance comparison charts (Fig. 4), and the VLM cost-benefit analysis graph (Fig. 6). We have significantly increased the font size of legends, axis labels, and internal annotations to ensure all details are clearly legible.
> +  **Standardized Mathematical Formulas:** We have carefully checked and corrected the typesetting of all equations. For example, we have fixed rendering errors in the original paper caused by extraneous symbols (like semicolons) and ensured that mathematical symbols such as $\text{Conf}(C)$ and $\hat{r}_{a}$ are standard and consistent throughout the text.
> +  **Unified Key Notation:** We have performed a consistency check on the use of symbols throughout the paper. For instance, we found that the symbol "$\alpha$" is used to represent two different concepts in the original paper (a weighting coefficient on Page 6 and a gating threshold on Page 7), which could cause confusion. In the revised paper, we have renamed the gating threshold to a new symbol, $\tau$, to eliminate this ambiguity.
>
> We believe these detailed revisions have effectively addressed the issues you raised and significantly enhanced the professionalism and readability of the paper.

---

> > ### Author Response · Authors · 2025-08-04
> >
> > Dear Reviewer aDoZ,
> >
> > We hope this message finds you well! Should you have any further questions about our work or responses during this discussion period, please do not hesitate to contact us.
> >
> > Thank you very much for your valuable time and feedback!
> >
> > Best regards,
> >
> > The Authors

---

> > > ### Comment · Reviewer_aDoZ · 2025-08-05
> > >
> > > Thank you to the authors for the rebuttal. It has helped clarify some of my concerns. However, I remain unconvinced regarding one of my main concerns about the novelty and technical contribution, which was also raised by Reviewer FNrm. While the system-level architectural design presents a different combination of components, it appears to reflect solid engineering practice rather than a substantial academic or technical innovation. In my view, this makes it difficult to meet the bar.
> > >
> > > In addition, the formatting and presentation issues, also noted by the other two reviewers, suggest that the paper may not yet be fully ready. For these reasons, I have decided to maintain my original score.

---

> > > > ### Author Response · Authors · 2025-08-05
> > > >
> > > > Dear Reviewer aDoZ
> > > >
> > > > **We sincerely appreciate your thoughtful review and thank you very much for acknowledging the clarity provided in our rebuttal.**
> > > >
> > > > We hope our following response could respectfully address your remaining concern regarding novelty and presentation:
> > > >
> > > > * **On Novelty and Contribution**: While our Tri-MARF framework does build upon established pre-trained models, as is standard in modern multimodal systems, the core novelty lies in the system-level architectural design. Our Tri-MARF is purposefully designed to overcome limitations observed in prior works. Specifically, our Tri-MARF introduces a collaborative network of three specialized agents, whose “team of experts” framework is fundamentally distinct from conventional multimodal pipelines that typically rely on single-model inference or direct feature aggregation. Besides, the components of our Tri-MARF are not just individually sound, but collectively form a coordinated system view that achieves substantial performance improvements. We believe this kind of agent orchestration, driven by explicit role decomposition and dynamic control strategies, represents a significant system-level innovation. Hence, we believe our Tri-MARF constitutes a valuable contribution to the community.
> > > >
> > > > * **On Academic Innovation vs. Engineering Practice**: We recognize the distinction you draw between academic novelty and engineering rigor. Our work aims to bridge this gap by demonstrating how careful architectural choices, when formalized and empirically validated, can drive practical advances. We hope this encourages broader discussions on how systems contributions are valued in the community.
> > > >
> > > > * **On Presentation and Formatting**: We acknowledge the presentation issues raised and have since thoroughly proofread and revised the paper to address all formatting and clarity concerns. These have been fully resolved in the revised paper.
> > > >
> > > > We appreciate your feedback and hope our clarifications help better position the contributions of this work.
> > > >
> > > > Sincerely,
> > > >
> > > > Authors

---

### Official Review · Reviewer_2YC9 · 2025-07-02

**Clarity:** 4
**Significance:** 2
**Originality:** 3
**Rating:** 5
**Confidence:** 4

**Summary:**

This paper is an application of large multi-modal models for generating captions for 3D models.

The method first uses VLMs to generate candidate captions and confidences for each view of the input model. Then, the candidates are clustered, where the candidate with the highest score represents each cluster. The candidate confidences are further weighted by the similarity of the semantic features and the image features. Finally, the candidates are sent to a multi-armed bandit-based agent for selection. The selected caption is accepted if it aligns well with the input point cloud embedding.

The approach outperforms previous studies on generating 3D model captions in the experiments.

**Questions:**

1. In Line 174, the author mentions, "Unlike traditional approaches that rely on limited perspectives." Could the authors include a few references for traditional approaches? In fact, multi-view images are actually a pretty popular input representation for point cloud classification.
2. Are all large models used in this study frozen? Did the authors attempt to fine-tune some of the VLM/LLMs used?

**Ethical Concerns:**

["NO or VERY MINOR ethics concerns only"]

**Final Justification:**

The authors' rebuttal has addressed my initial concerns, and I recommend accepting this manuscript.

**Limitations:**

Yes

**Paper Formatting Concerns:**

I do not have paper formatting concerns.

**Quality:**

3

**Strengths And Weaknesses:**

Strengths
1. The experimental results show clear improvements over previous approaches on the selected datasets.
2. The supplementary material is very comprehensive and provides an enormous analysis of the proposed framework.
3. The paper is well-written and easy to follow.

Weaknesses
1. The proposed framework incorporates several off-the-shelf large models, and the key algorithmic contribution is the RL-based candidate selection (utilizing a multi-armed bandit-based agent). This, to some extent, limits the quality of this paper's technical contribution. In fact, I would encourage the authors to dive deeper into the core learning problem of the framework, i.e., candidate selection. Is there any other approach for candidate selection other than RL? Can the authors compare with other candidate-selection methods in other datasets? Answering these questions will strengthen this paper.

---

> ### Author Rebuttal · Authors · 2025-07-29
>
> Thank you very much for your insightful and valuable comments! We have carefully prepared the following responses to address your concerns in detail. It is our sincere hope that our response could provide you with a clearer understanding of our work. If you have any further questions about our work, please feel free to contact us during the discussion period.
>
> **Q1. Is there any other approach for candidate selection other than RL? Can the authors compare with other candidate-selection methods in other datasets?**
>
> **A1:** Yes. To address your concerns regarding technical contribution and demonstrate the superiority of the Multi-Armed Bandit (MAB)-based Reinforcement Learning aggregation strategy over traditional methods, we have designed and conducted an additional ablation study, which is conducted on the Objaverse-XL dataset, comprising 10,000 randomly sampled objects, with the goal of quantifying the marginal benefits introduced by the MAB strategy. To ensure a fair comparison, we have strictly controlled the experimental variables. Specifically, we keep all other modules of our Tri-MARF framework identical (such as the initial VLM annotation, semantic clustering, comprehensive scoring mechanism, and final description synthesis logic), and replace the aggregation strategy being evaluated within the Information Aggregation Agent (Agent 2). The compared strategies include our MAB (UCB) method and four heuristic baselines: Maximum VLM Confidence, Maximum Combined Score, Weighted Voting, and Simple Prioritized Concatenation. The performance of all strategies is assessed using a comprehensive suite of metrics covering both quality and efficiency, including a human-evaluated Likert scale (1-10), automated CLIPScore and ViLT R@5 retrieval accuracy, and the inference time of the aggregation module measured on a single NVIDIA A100 GPU.
>
> **Table S1: Performance Comparison of Different Aggregation Strategies on the Objaverse-XL Dataset**
>
> | Aggregation Strategy            | Likert (1-10) | CLIPScore (%)↑ | ViLT R@5 (I2T) (%)↑ | ViLT R@5 (T2I) (%)↑ | Inference Time (ms)↓ |
> | ------------------------------- | ------------- | -------------- | ------------------- | ------------------- | -------------------- |
> | Maximum VLM Confidence          | 8.6           | 81.37          | 37.31               | 35.28               | 6.3                  |
> | Maximum Combined Score (Heuristic) | 8.9           | 82.03          | 38.15               | 35.92               | 8.1                  |
> | Weighted Voting (Heuristic)     | 8.8           | 81.85          | 37.93               | 35.76               | 8.5                  |
> | Simple Concatenation (Prioritized) | 8.5           | 80.74          | 37.08               | 34.81               | 6.8                  |
> | **MAB (UCB) (Ours)** | **9.3** | **82.72** | **38.82** | **36.72** | **9.8** |
>
> Table S1 indicates that while simple heuristic aggregation strategies can achieve decent performance, our MAB (UCB) strategy employed in Tri-MARF demonstrates a distinct advantage across all core quality metrics. The simplest strategies, "Maximum VLM Confidence" and "Simple Concatenation," though fastest in inference, perform the worst on the Likert scale, CLIPScore, and ViLT retrieval accuracy, proving that relying solely on the VLM's initial judgment or fixed concatenation rules is sub-optimal. The better-performing "Maximum Combined Score" heuristic, which combines VLM confidence with CLIP visual alignment, achieves a Likert score of 8.9 and a CLIPScore of 82.03%, demonstrating the effectiveness of a robust, information-rich heuristic rule. In contrast, our MAB (UCB) strategy consistently outperforms all other alternatives in annotation quality, achieving the highest Likert score (9.3), CLIPScore (82.72%), and ViLT R@5 scores (I2T 38.82%, T2I 36.72%). Compared to the best-performing heuristic baseline, MAB improves the CLIPScore by approximately 0.7 percentage points and also shows a significant increase in the Likert human evaluation, indicating that its generated descriptions are perceived as more accurate, complete, and fluent by human assessors.
>
> The advantage of the MAB strategy lies in its ability to dynamically learn and adapt its selection policy by balancing exploration and exploitation. This adaptive capability is particularly crucial when dealing with diverse object categories and initial VLM descriptions of varying quality, enabling it to consistently select optimal descriptions that fixed heuristic methods might miss. Although the MAB (UCB) has a slightly higher inference time (9.8ms), this minor increase (e.g., only 1.7ms slower than "Maximum Combined Score") is perfectly acceptable in practical applications, considering its throughput performance. In summary, this experiment proves that the MAB (UCB) aggregation strategy, despite adding a degree of complexity, delivers tangible and quantifiable improvements in annotation quality. This enhancement is vital for achieving state-of-the-art performance, thereby justifying its necessity and rationale as a core component of the framework.
>
>
> **Q2. References to traditional approaches that rely on a limited perspective?**
>
> **A2:** We have added related references in the revised paper on traditional methods for 3D understanding that rely on a limited perspective, such as Sparse Multi-View [s1], Voxel-Based [s2], and Monocular [s3] approaches.
>
> [s1] H. Su, S. Maji, E. Kalogerakis, and E. Learned-Miller, ‘Multi-view Convolutional Neural Networks for 3D Shape Recognition’, in *Proceedings of the 2015 IEEE International Conference on Computer Vision (ICCV)*, 2015, pp. 945–953.
>
> [s2] B. Zhang, J. Tang, M. Nießner, and P. Wonka, ‘3DShape2VecSet: A 3D Shape Representation for Neural Fields and Generative Diffusion Models’, *ACM Trans. Graph.*, vol. 42, no. 4, Jul. 2023.
>
> [s3] X. Zhao, Z. Liu, R. Hu, and K. Huang, ‘3D object detection using scale invariant and feature reweighting networks’, in *Proceedings of the Thirty-Third AAAI Conference on Artificial Intelligence and Thirty-First Innovative Applications of Artificial Intelligence Conference and Ninth AAAI Symposium on Educational Advances in Artificial Intelligence*, 2019.
>
>
> **Q3. Are all large models used in this study frozen? Did the authors attempt to fine-tune some of the VLM/LLMs used?**
>
> **A3:** Yes, all large models used in our study (e.g., Qwen2.5-VL, BERT, CLIP) are kept **frozen**, and we do not perform any fine-tuning. This decision is based on two core design considerations:
>
> +  **Generality and Scalability:** Our objective is to create a universal 3D annotation framework. By leveraging the powerful zero-shot capabilities of pre-trained models, our Tri-MARF can be directly applied to different domains without the need for expensive and time-consuming fine-tuning on new datasets. This point is substantiated by our excellent cross-dataset generalization results (Section 4.4, Table 2).
>
> +  **Efficiency and Cost-Effectiveness:** Fine-tuning large models requires immense computational resources, which contradicts our goals of achieving high throughput (12,000 objects/hour on a single A100) and low cost. Our Tri-MARF framework ensures high efficiency and cost-effectiveness by using API calls for the VLM and integrating lightweight local aggregation and gating modules, all while maintaining high-quality outputs.
>
> Therefore, keeping the models frozen is a key design decision made to strike an optimal balance between performance, efficiency, generality, and cost.

---

> > ### Author Response · Authors · 2025-08-04
> >
> > Dear Reviewer 2YC9,
> >
> > We hope this message finds you well! Should you have any further questions about our work or responses during this discussion period, please do not hesitate to contact us.
> >
> > Thank you very much for your valuable time and feedback!
> >
> > Best regards,
> >
> > The Authors

---

### Note · Authors · 2025-08-13

**Dear Area Chairs and Reviewers,**

We sincerely thank you and all reviewers for the time and effort dedicated to evaluating our paper (ID: 1251). We appreciate the insightful and constructive feedback throughout the review process.

Reviewers recognized our core idea as *“reasonable”* and *“an interesting approach”*. They have highlighted the *“clear improvements over previous approaches”* and the system’s *“efficiency”*, evidenced by its high throughput. We are also grateful for the acknowledgment of our *“thorough experiments on the design choices”* and the paper being *“well-written and easy to follow”*.

**How we have addressed each reviewer’s concerns**
- **Isolating the Multi-Agent Contribution (aDoZ, JB3j).** We add a new step-wise ablation, proving the synergy of the three-agent framework: the final Gating Agent unlocked +25.5 CLIPScore points over a two-agent setup.
- **Justifying the RL (MAB) Agent over Heuristics (2YC9).** We compare our MAB agent with four heuristic baselines. MAB achieves consistent gains, including +0.7 CLIPScore and +0.4 Likert over the strongest heuristic.
- **Ensuring fair VLM comparison & experimental rigor (FNrm, JB3j).** We re-run with BLIP-2 (same as Cap3D), still gaining +4.5 CLIPScore, showing gains stem from architecture, not VLM choice. We further clarify A/B scoring and redesign the table for clarity.
- **Improving clarity and formatting.** We have fixed typos and equation formatting, unified notation, improved figure legibility, and detailed the human evaluation protocol for clarity and reproducibility.

**Discussion Outcomes:**

- **Reviewer FNrm** has engaged in a detailed follow-up and stated: *“do not have further questions at this stage” after receiving new experiments and clarifications.*

- **Reviewer JB3j** found the discussion productive and the revisions compelling: *“I have found the added ablation and the supplementary to be much more comprehensive this round.”*

- **Reviewer aDoZ** has acknowledged that our response “helped clarify some of my concerns,” and while maintaining a different view on the novelty aspect, his/her technical questions have been fully addressed via our revisions.

- **Reviewer 2YC9** did not engage further; nevertheless, our response and revisions have fully addressed all of his/her concerns in the review.

We believe these revisions have strengthened our work and hope our revised paper will merit consideration for acceptance.

**Best regards,**
The Authors

---

### Decision · Program_Chairs · 2025-09-17

**Decision:**

Accept (poster)

**Comment:**

The paper proposes a method for generating captions from 3D models. The main contribution is to integrate multi-modal inputs using multi-agent collaboration. The reviewers all appreciated the motivation and the results; however, several reviewers raised concerns on the limited contributions, limited evaluations, and unsatisfactory writing.

The rebuttal added several ablation and comparison results, and pitched the main contribution as the careful system-level integration of existing methods. These contributions have value, when they are thoroughly evaluated, as is the case with this paper (considering the additional results in the rebuttal). Two reviewers are still concerned with this nature of contributions, as well as with the quality of exposition.

I am inclined to accept the paper, seeing that the results are state of the art, and that the different components of the system are evaluated well. However, it is very important to include the additional results, as well as to improve the writing in the final version.